# MyosinA is a druggable target in the widespread protozoan parasite *Toxoplasma gondii*

Anne Kelsen[1]☯, Robyn S. Kent[1]☯, Anne K. Snyder[1], Eddie Wehri[2], Stephen J. Bishop[3], Rachel V. Stadler[1], Cameron Powell[4], Bruno Martorelli di Genova[1], Pramod K. Rompikuntal[1], Martin J. Boulanger[4], David M. Warshaw[5], Nicholas J. Westwood[3], Julia Schaletzky[2], Gary E. Ward[1] *

**1** Department of Microbiology and Molecular Genetics, University of Vermont Larner College of Medicine, Burlington, Vermont, United States of America, **2** Center for Emerging and Neglected Diseases, University of California Berkeley, California, United States of America, **3** School of Chemistry and Biomedical Sciences Research Complex, University of St. Andrews and EaStCHEM, St Andrews, Fife, Scotland, United Kingdom, **4** Department of Biochemistry and Microbiology, University of Victoria, Victoria, BC, Canada, **5** Department of Molecular Physiology and Biophysics, University of Vermont Larner College of Medicine, Burlington, Vermont, United States of America

☯ These authors contributed equally to this work.
* Gary.Ward@uvm.edu

**Data Availability Statement:** All relevant data are within the paper and its Supporting Information files.

## Abstract

*Toxoplasma gondii* is a widespread apicomplexan parasite that can cause severe disease in its human hosts. The ability of *T. gondii* and other apicomplexan parasites to invade into, egress from, and move between cells of the hosts they infect is critical to parasite virulence and disease progression. An unusual and highly conserved parasite myosin motor (TgMyoA) plays a central role in *T. gondii* motility. The goal of this work was to determine whether the parasite's motility and lytic cycle can be disrupted through pharmacological inhibition of TgMyoA, as an approach to altering disease progression in vivo. To this end, we first sought to identify inhibitors of TgMyoA by screening a collection of 50,000 structurally diverse small molecules for inhibitors of the recombinant motor's actin-activated ATPase activity. The top hit to emerge from the screen, KNX-002, inhibited TgMyoA with little to no effect on any of the vertebrate myosins tested. KNX-002 was also active against parasites, inhibiting parasite motility and growth in culture in a dose-dependent manner. We used chemical mutagenesis, selection in KNX-002, and targeted sequencing to identify a mutation in TgMyoA (T130A) that renders the recombinant motor less sensitive to compound. Compared to wild-type parasites, parasites expressing the T130A mutation showed reduced sensitivity to KNX-002 in motility and growth assays, confirming TgMyoA as a biologically relevant target of KNX-002. Finally, we present evidence that KNX-002 can slow disease progression in mice infected with wild-type parasites, but not parasites expressing the resistance-conferring TgMyoA T130A mutation. Taken together, these data demonstrate the specificity of KNX-002 for TgMyoA, both in vitro and in vivo, and validate TgMyoA as a druggable target in infections with *T. gondii*. Since TgMyoA is essential for virulence, conserved in apicomplexan parasites, and distinctly different from the myosins found in

**Funding:** This work was supported by the National Institutes of Health (AI139201 and AI137767 to GEW, each including salary support; GM141743 to DMW, including salary support; F31AI145214 to RVS, including predoctoral fellowship stipend support; and T32AI055402 to GEW, including predoctoral fellowship stipend support for AKS). The work was also supported by the Canadian Institutes of Health Research (148596 to MJB), the Canada Research Chair program (to MJB, salary support) and the American Heart Association (20POST35220017 to RSK, including postdoctoral fellowship stipend support). The funders had no role in study design, data collection and analysis, decision to publish, or preparation of the manuscript.

**Competing interests:** The authors have declared that no competing interests exist.

**Abbreviations:** CI, confidence interval; DSF, differential scanning fluorimetry; FBS, fetal bovine serum; HCM, hypertrophic cardiomyopathy; HFF, human foreskin fibroblast; RFU, relative fluorescence unit; RLU, relative luminescence unit; SAR, structure-activity relationship; SEC, size-exclusion chromatography; SMM, smooth muscle myosin.

humans, pharmacological inhibition of MyoA offers a promising new approach to treating the devastating diseases caused by *T. gondii* and other apicomplexan parasites.

## Introduction

Nearly one third of the world's population is or has been infected with the apicomplexan parasite, *Toxoplasma gondii*. Although most infections are subclinical, acute toxoplasmosis can have severe consequences in neonates and immunocompromised individuals. Congenital infection can lead to spontaneous abortion or stillbirth, and even children born with subclinical infection frequently experience sequelae later in life including neurological damage and vision impairment [1,2]. Among the immunocompromised, toxoplasmic encephalitis is a particularly significant risk in AIDS patients who are unaware of their HIV status [3–5] and in the approximately 50% of HIV-infected individuals worldwide who do not have access to antiretroviral therapy [6,7]. In individuals with AIDS, adverse effects of the currently available drugs to suppress toxoplasmosis [8] cause the discontinuation of treatment in up to 40% of patients [4,9–11], highlighting the need for new, better-tolerated drugs to reduce toxoplasmosis-related morbidity and mortality.

Within the brain and other tissues of a patient suffering from active toxoplasmosis, the tachyzoite stage of the parasite uses a unique form of substrate-dependent "gliding" motility to invade into and egress from the cells of its host, to spread throughout the patient's tissues, and to migrate across biological barriers [12–15]. *T. gondii* tachyzoites move in the extracellular matrix along flattened corkscrew-like trajectories, with regularly repeating periods of acceleration and deceleration [16]. The mechanisms underlying gliding motility are thought to be conserved among apicomplexan parasites, including *Plasmodium* spp. (the causative agents of malaria) and *Cryptosporidium* spp. (which cause severe diarrheal disease). Motility is driven, at least in part, by an unusual class XIVa myosin motor protein, myosinA (MyoA), found only in apicomplexan parasites and a few ciliates. In *T. gondii*, this motor consists of the TgMyoA heavy chain and 2 associated light chains, TgMLC1 and either TgELC1 or TgELC2 [17–19]. TgMyoA is a single-headed motor with a very low duty ratio (i.e., it remains strongly bound to actin for only approximately 1% of its catalytic cycle) yet moves actin filaments at the same speed as skeletal muscle myosin, a double-headed motor with a 5-fold higher duty ratio [20,21]. TgMyoA lacks a conventional tail, as well as several residues conserved in other myosins that normally function to regulate actomyosin activity [22,23] and employs atypical strategies for chemomechanical coupling and force transduction [24,25].

Depletion of TgMyoA results in severely reduced parasite motility, host cell invasion, and host cell egress [26–29]. As a result, parasites lacking TgMyoA are avirulent in an animal model of infection [28]. Despite its importance in the *T. gondii* life cycle, we know little about how TgMyoA generates the complex pattern of parasite motility seen in 3D or how TgMyoA function contributes to disease pathogenesis in an infected host. Because the parasite can compensate for the loss or reduced expression of proteins important to its life cycle [30–32], small-molecule inhibitors of TgMyoA would serve as valuable complementary tools for determining how different aspects of motor function contribute to parasite motility and the role played by TgMyoA in parasite dissemination and virulence. Specific inhibitors are also necessary to establish TgMyoA as a "druggable" target, i.e., a protein whose activity is amenable to inhibition by small molecules, and to test whether pharmacological inhibition of TgMyoA activity in vivo alters the course of an infection.

We describe here the identification and characterization of KNX-002, the first specific inhibitor of an apicomplexan class XIVa myosin. KNX-002 inhibits TgMyoA motor activity,

parasite motility, and parasite growth in culture in a dose-dependent manner, with little or no effect on a variety of other vertebrate myosins. On-target activity in parasites was confirmed by identifying a mutation in TgMyoA that rescues parasite motility and growth in the presence of the compound. The availability of a specific inhibitor of TgMyoA, together with isogenic parasite lines that show differential sensitivity to the compound, enabled us to undertake the first direct test of whether pharmacological inhibition of MyoA can be used to alter the progression of disease caused by an apicomplexan parasite.

## Results

### High-throughput screening identifies a novel inhibitor of TgMyoA

To identify inhibitors of TgMyoA motor activity, we used our previously described method for producing large amounts of functional motor, in which the TgMyoA heavy chain is co-expressed in insect cells with its 2 light chains, TgMLC1 and TgELC1, and a myosin co-chaperone protein [33]. We developed a miniaturized coupled enzyme assay to measure actin-dependent ATPase activity of the purified recombinant motor and screened 50,000 compounds from the compound library at Cytokinetics Inc. for inhibitors of this ATPase activity. Hit follow-up and characterization included dose-response analysis using resupplied compound that was determined to be >95% pure by LC/MS analysis and control assays demonstrating the compound was inactive against an unrelated ATPase, hexokinase. The most potent hit was 1-(4-methoxyphenyl)-N-((3-(thiophen-2-yl)-1H-pyrazol-4-yl)methyl)cyclopropan-1-amine (Fig 1A, inset), subsequently named KNX-002. KNX-002 inhibits TgMyoA ATPase activity in a dose-dependent manner with an $IC_{50}$ of 2.8 μM (95% confidence interval (CI) 2.4 to 3.2 μM; Fig 1A). In contrast, KNX-002 has little to no detectable effect on the actin-activated ATPase activity of bovine cardiac myosin, chicken gizzard smooth muscle myosin (SMM), rabbit fast skeletal muscle myosin, or bovine slow skeletal muscle myosin, up to the highest concentrations tested (40 μM; Figs 1 and S1). These data identify KNX-002 as a potentially specific inhibitor of *T. gondii* class XIVa myosin activity.

### KNX-002 inhibits the growth of *T. gondii* in cell culture, with no detectable toxicity to human cells

The 3 lipid bilayers that comprise the pellicle of *T. gondii*, i.e., the plasma membrane and subjacent double bilayer of the inner membrane complex [34,35], represent a potential barrier to the penetration of externally applied drugs. To determine if KNX-002 can access and inhibit the motor in live parasites, we determined the compound's effect on parasite expansion in culture in a modified plate-based growth assay (see Methods and ref. [36]). KNX-002 does indeed inhibit parasite growth, in a dose-dependent manner (Fig 2A), with an $IC_{50}$ of 16.2 μM (95% CI 13.0 to 20.5 μM; Fig 2B).

Parasite growth in culture involves repeated lytic cycles comprised of host cell invasion, intracellular replication, lysis of and egress from the infected cell, and migration to new host cells. While invasion, egress, and migration are all TgMyoA-dependent processes, intracellular replication does not require TgMyoA [26,28]. To test whether any portion of the observed growth inhibition in culture is due to off-target effect(s) of KNX-002 on parasite replication, we scored the number of parasites per parasitophorous vacuole over time during the lytic cycle. The compound had no detectable effect on parasite replication in this assay (S2 Fig).

To exclude the possibility that the effect of KNX-002 on parasite growth was due to a deleterious effect on the host cells, we also tested the effect of the compound on human foreskin fibroblasts (the host cells used in the growth assays) and human HepG2 liver cells (sensitive

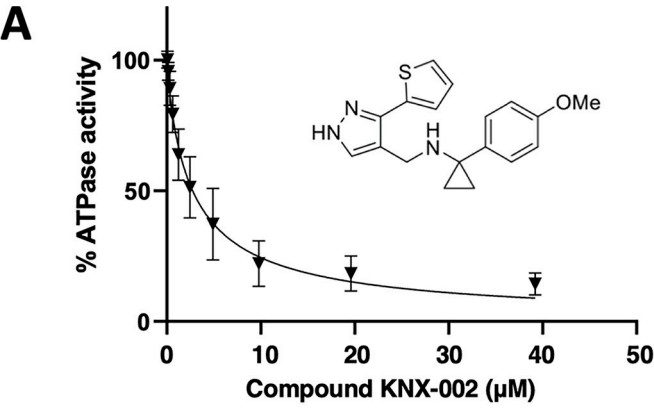

**A**

**B**

| Test enzyme | IC$_{50}$ | 95% C.I. |
|---|---|---|
| TgMyoA | 2.8 μM | 2.4-3.2 μM |
| Chicken SMM | >39.2 μM | NA |
| Bovine cardiac myofibrils (pCa75) | >39.2 μM | NA |
| Rabbit fast skeletal myofibrils (pCa75) | >39.2 μM | NA |
| Bovine slow skeletal myofibrils (pCa75) | >39.2 μM | NA |
| Bovine slow skeletal myofibrils (pCa25) | >39.2 μM | NA |
| Hexokinase | >39.2 μM | NA |

**Fig 1. KNX-002 inhibits the actin-activated ATPase activity of TgMyoA but not vertebrate myosins. (A)** The effect of various concentrations of KNX-002 on TgMyoA ATPase activity, normalized to the activity with an equivalent amount of DMSO (vehicle) only. Inset shows the structure of KNX-002. Each data point represents mean ATPase activity of 3 different batches of KNX-002 ± SEM. **(B)** The inhibitory effect of KNX-002 on TgMyoA and various vertebrate myosins (IC$_{50}$ and 95% CI). The dose-response curves for the individual vertebrate myosins are shown in S1 Fig and the measurements underlying the data summarized in this figure can be found in S1 Data. CI, confidence interval.

indicators of cytotoxicity and genotoxicity [37]) in 72-h multiplexed cytotoxicity and cellular viability assays. KNX-002 showed no detectable deleterious effect on either cell line in either assay at the highest concentration tested (80 μM; Fig 2C).

## Structure-activity relationship (SAR) analysis

Next, we used the ATPase and parasite growth assays to carry out a preliminary structure-activity relationship (SAR) study, with the goal of determining which parts of KNX-002 are important for its biological activity. We began by modifying the *p*-OMe substituted aryl ring in KNX-002 (colored red in KNX-002 structure, Fig 3). Replacement of this structural unit in full by a hydrogen atom (in **VEST1**) confirmed the importance of this part of the molecule for biological activity (Figs 3 and S3 and S4). In terms of the preferred substituent on this aryl ring, inhibition of both in vitro ATPase activity and parasite growth were retained when the *p*-OMe substituent in KNX-002 was replaced by *p*-Cl (**VEST2**), and partially retained in analogs

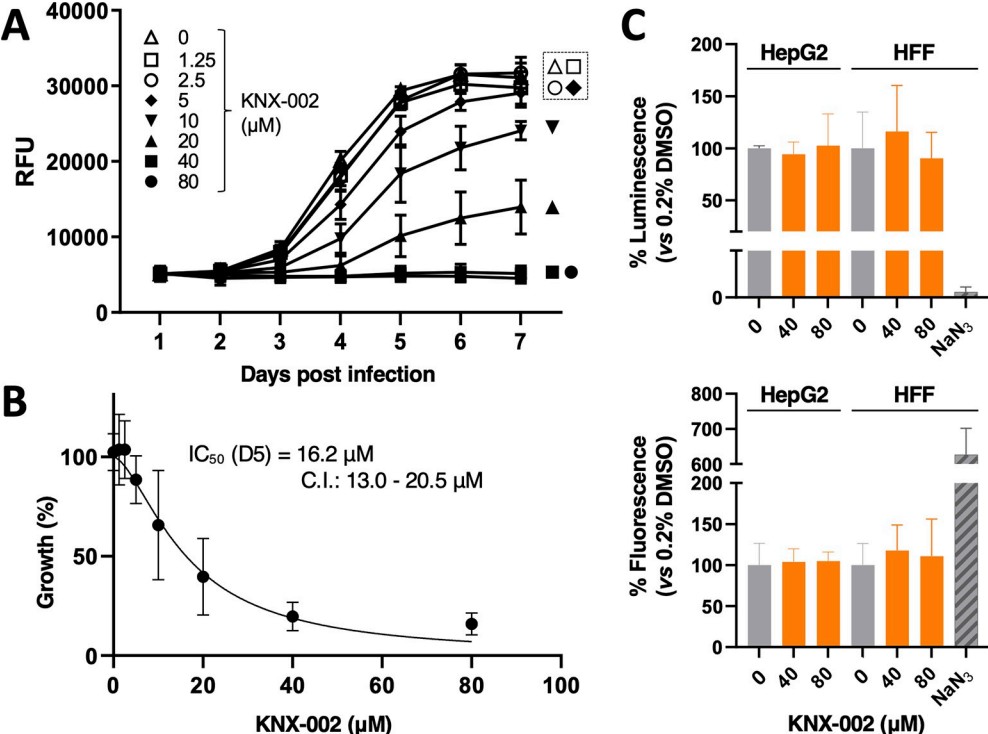

**Fig 2. KNX-002 inhibits the growth of *T. gondii* in culture, with no detectable host cell toxicity. (A)** tdTomato-expressing parasites were preincubated with various concentrations of KNX-002 for 5 min (0 = DMSO vehicle only) and then added to HFF cells on a 384-well plate. Fluorescence was measured daily to quantify parasite growth, in the continued presence of compound, over the next 7 days. RFU = relative fluorescence units. The data shown are the mean ± SEM from 2 independent biological replicates, each consisting of 3 technical replicates at all time points. **(B)** The data from day 5 (D5) of the growth assay shown in panel (A) were used to calculate the $IC_{50}$ of KNX-002 for parasite growth (16.2 μM, 95% CI = 13.0 to 20.5). See Fig 5A for a repeat of these growth assays and $IC_{50}$ calculations. **(C)** Subconfluent HepG2 and HFF cells were cultured in the presence of 0 (DMSO vehicle only), 40, or 80 μM KNX-002 for 72 h. Cell viability/growth was measured using Promega CellTiter-Glo (top panel) and toxicity was measured using Promega CellTox green (bottom panel), and 0.1% w/v $NaN_3$ served as a positive control for cytotoxicity. Results are plotted relative to the DMSO controls. Bars show the mean of 3 independent experiments ± SEM. KNX-002 shows no evidence of growth inhibition or cytotoxicity in either cell type at either 40 or 80 μM. The measurements underlying the data plotted in this figure can be found in S2 Data. HFF, human foreskin fibroblast.

containing *p*-Me (**VEST3**) or *m,p*-dichloro (**VEST4**) substituents. **VEST5**, an analog in which the *p*-OMe in KNX-002 was replaced by a hydrogen, was significantly less active than KNX-002 in the purified protein assay but a strong inhibitor of parasite growth. **VEST5** was not toxic to host cells, extracellular parasites, or intracellular parasites (S5 and S6 Figs); further studies will be required to understand the activity of this analog in the parasite growth assay. The mild to moderate toxicity levels shown by **VEST2** (*p*-Cl) against HepG2 and human foreskin fibroblast (HFF) cells, compared to the lack of toxicity in either cell line demonstrated by KNX-002, led to the preliminary conclusion that the *p*-OMe aryl substituent present in the starting KNX-002 analogue was preferred.

A series of additional analogs containing the *p*-OMe aryl substituent were also assessed. Substitution of the thiophene ring in KNX-002 for a phenyl ring (in **VEST6,** highlighted in blue, Fig 3) led to a drop in activity in both the in vitro ATPase and growth inhibition assays. However, the retention of some biological activity by **VEST6** may provide a way to assess rapidly the importance of additional substitution in this region of the molecule in future studies through the incorporation of substituents on the new phenyl ring. No tolerance for

**Fig 3. Preliminary SAR analysis.** The structures of KNX-002 and 15 analogs (VEST1-15) are shown at the top. The table summarizes the activity of each of the compounds in assays measuring recombinant TgMyoA ATPase activity, parasite growth, and mammalian cell toxicity; the data on which the table is based are shown in S3–S5 Figs. ATPase assay data correspond to relative luminescence units $\times 10^{-6}$ (mean from 3 independent replicates +/-SEM) in the presence of 20 μM compound (S3 Fig). Green = less than 20% inhibition, yellow = 20%–40% inhibition; orange = 45%–70% inhibition; red = greater than 70% inhibition compared to vehicle (DMSO) controls. Growth assay data correspond to calculated $IC_{50}$ and 95% CI from a 5 to 6 day growth assay (S4 Fig). Red = $IC_{50}$ less than 20 μM; orange = $IC_{50}$ of 20–35 μM; yellow = $IC_{50}$ of 35–50 μM; green = no detectable growth inhibition relative to vehicle (DMSO) controls. Toxicity assay entries summarize the results from 72-h CellTox green toxicity and CellTiter-Glo viability assays on both human foreskin fibroblasts and HepG2 cells. Thresholds for mild and moderate toxicity and loss of viability compared to vehicle (DMSO) controls are indicated by the colored bars on the right-hand side of S5 Fig. CI, confidence interval; SAR, structure-activity relationship.

replacement of the pyrazole (highlighted in green) and thiophene (blue) rings in KNX-002 was observed, with analogs containing only a phenyl, thiazole, or furan substituent, instead of the thiophene-linked pyrazole, all being inactive in both assays (**VEST7**, **VEST8**, and **VEST9**). Substitution of a pyrrole into KNX-002 to give **VEST10** led to loss of activity in the in vitro

assay but retention of some activity in the growth inhibition assay. Finally, the importance of the cyclopropyl group in KNX-002 (highlighted in purple, Fig 3) was confirmed by the lack of biological activity in either assay of the *des*-cyclopropyl analog **VEST11**. The growth inhibition activity associated with **VEST5** was also shown to require the cyclopropyl group as analogs **VEST12** ($CH_2$) and **VEST13** ($CMe_2$) were both less active in the growth assay. Attempts to incorporate a stereogenic center through the replacement of the cyclopropyl group with a single methyl substituent led to analogs with decreased biological activity and mild cytotoxicity against HFF cells (**VEST14** and **VEST15**).

These preliminary studies have identified regions of KNX-002 important for its biological activity, which will inform subsequent efforts at compound optimization (see Discussion). Since no analogs with higher potency than KNX-002 were identified in this first round of SAR analysis, all experiments reported below were carried out using KNX-002.

## KNX-002 inhibits parasite motility

To test whether KNX-002 inhibits parasite motility, we first used a standard immunofluorescence-based assay that visualizes the protein "trails" left behind the parasites as they glide on a glass coverslip (e.g., [38,39]). Untreated parasites deposit a variety of helical, meandering, and large diameter circular trails in this qualitative 2D assay (Fig 4A). Treatment with 10 μM KNX-002 results in a different motility pattern, consisting almost entirely of small diameter circles, and at 25 μM KNX-002 most parasites do not move at all (Fig 4A).

To more quantitatively evaluate differences in parasite motility in the presence of KNX-002, we used a 3D motility assay that automatically analyzes hundreds to thousands of trajectories in a single experiment, providing robust statistical comparisons between populations of parasites [16,21]. KNX-002 treatment decreases the fraction of the parasite population moving in the 3D assay in a dose-dependent manner (Fig 4B and 4C; $IC_{50}$ = 6.2 μM, 95% CI 4.1 to 9.4 μM). Of those parasites that move, KNX-002 treatment also reduces their average displacement and speed (S7 Fig).

## Identification of a mutation in TgMyoA that confers partial resistance to KNX-002

The results presented thus far show that KNX-002 is an inhibitor of TgMyoA activity, parasite growth in culture, and parasite motility. The data do not, however, directly link the compound's effects on parasite motility and growth to inhibition of TgMyoA function; it remains possible that the inhibition of parasite motility and growth are due to off-target effect(s). As a first step towards validating the specificity of the compound, we chemically mutagenized parasites with N-ethyl-N-nitrosourea [40] and selected for parasites resistant to KNX-002 by growth in 40 μM KNX-002. Parasites that grew up after 10 to 14 days under selection were cloned and their sensitivity to KNX-002 determined using the plate-based growth assay. We isolated 26 such clones from 2 independent mutagenesis/selection experiments, 5 of which showed at least a 2.5-fold increase in $IC_{50}$ for KNX-002 (S8 Fig). cDNA was generated from the 5 clones, and each of the motor constituents (TgMyoA, TgMLC1, TgELC1, and TgELC2) was sequenced. Four of the five resistant clones contained no amino acid-altering mutations in the motor genes; the fifth (clone R3) contained a single A to G point mutation in the coding sequence of TgMyoA that changes threonine130 to alanine (T130A). This clone showed a 2.9-fold increase in $IC_{50}$ for KNX-002 in the growth assay (Fig 5A and 5B), from 14.9 μM (wild type; 95% CI = 10.2 to 22.6 μM) to 43.4 μM (T130A mutant; 95% CI = 37.0 to 51.2 μM). T130 is highly conserved in eukaryotic myosins (Fig 5C) and located on the first of 7 β-strands comprising the transducer subdomain (Fig 5D). The residues immediately flanking T130 are also

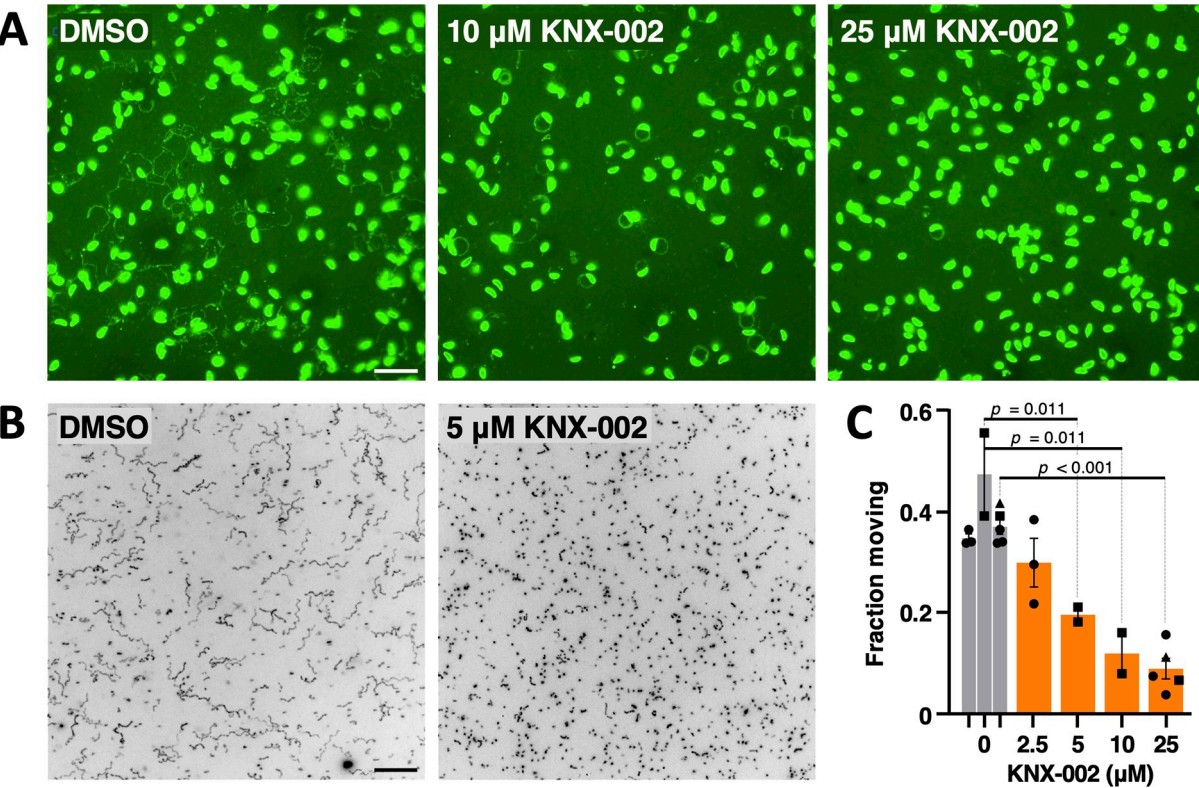

**Fig 4. KNX-002 inhibits the motility of *T. gondii* tachyzoites. (A)** Representative images from a 2D trail assay of parasites treated with DMSO (vehicle) only, 10 µM KNX-002, or 25 µM KNX-002. The parasites and trails were visualized by indirect immunofluorescence using an antibody against TgSAG1 after 15 min of gliding on the coverslip. Scale bar = 20 µm. **(B)** Representative maximum intensity projections showing parasite trajectories during 60 s of motility in Matrigel in the absence (DMSO) or presence of 5 µM KNX-002. Scale bar = 40 µm. The grayscale images were inverted to provide clearer visualization of the trajectories. **(C)** Fraction of the parasite population moving during a 60-s assay in the presence of the indicated concentrations of KNX-002 ($IC_{50}$ = 6.2 µM, 95% CI = 4.1–9.4 µM). Each data point represents a single biological replicate consisting of 3 technical replicates. Sets of data captured on the same days, indicated by similar symbol shapes, were compared by Student's one-tailed paired *t* tests. The measurements underlying the data plotted in panel C can be found in S9 and S10 Data. CI, confidence interval.

well conserved between the MyoA homologs of other apicomplexan parasites, but diverge in the vertebrate myosins examined (Fig 5C).

To independently establish that the T130A mutation alters the sensitivity of TgMyoA to KNX-002, we expressed recombinant TgMyoA containing the T130A mutation and compared its motion-generating capacity to that of wild-type TgMyoA in an in vitro motility assay that measures the speed with which fluorescently labeled actin filaments are moved by myosin adsorbed to a glass coverslip [20,41]. In the absence of added compound, the wild-type motor moved 92.5 ± 1.2% of the filaments on the coverslip, with an average mean speed of 4.6 ± 0.3 µm/s (Fig 6A, top and middle panels, gray bars). The T130A mutation had no effect on the fraction of filaments moving in the assay but decreased the basal speed of filament sliding by 77 ± 3% compared to wild-type (Fig 6B, top and middle panels, gray bars). Accordingly, we tested whether the mutant was misfolded, prone to aggregation, or otherwise unstable by comparing its size-exclusion chromatography (SEC) elution profile and thermal stability, as measured by differential scanning fluorimetry (DSF), to that of the wild-type motor (S9 Fig; both proteins assayed in the near-rigor [SEC] and pre-powerstroke [DSF] states). No mutation-associated differences were seen in either assay, suggesting that the effects of the T130A

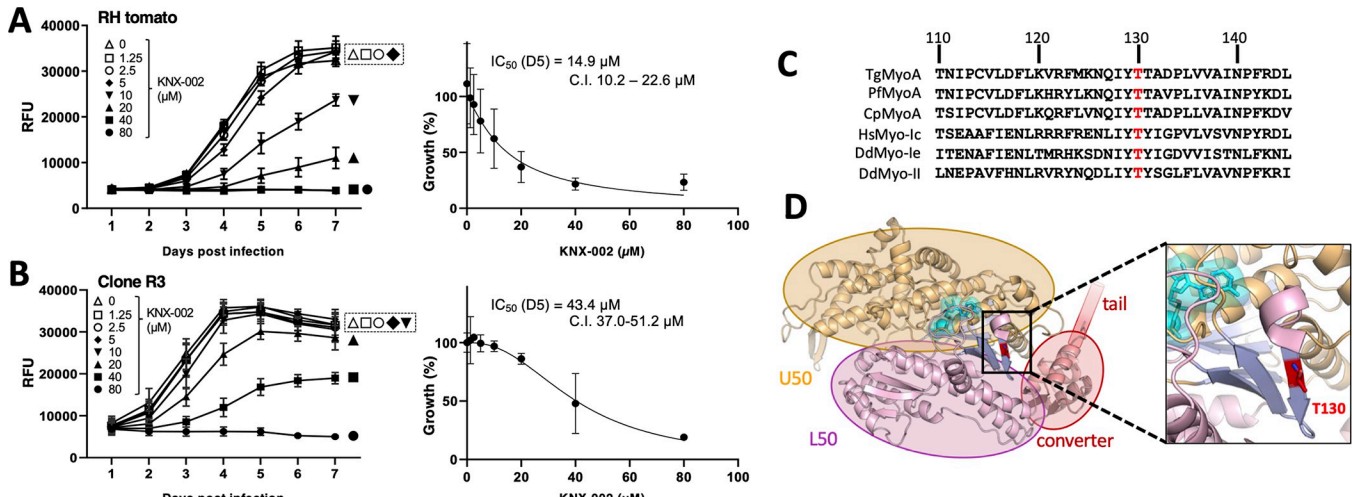

**Fig 5. Identification of a mutant parasite showing reduced sensitivity to KNX-002 in growth assays. (A, B)** Left panels: tdTomato-expressing wild-type parasites (A) or clone R3 (B), which was generated by mutagenesis and selection in KNX-002, were preincubated with various concentrations of KNX-002 for 5 min and then added to HFF cells on a 384-well plate. Fluorescence was measured daily over the next 7 days to quantify parasite growth in the continuing presence of compound. RFU = relative fluorescence units. The data shown are from 3 independent biological replicates, each consisting of 2 to 3 technical replicates at all time points. Right panels: the data from day 5 (D5) of the growth assays were used to calculate the $IC_{50}$ and 95% CI of KNX-002 for parasite growth. Vertical bars indicate SEM (left panels) or 95% CI (right panels). **(C)** Multiple sequence alignment showing the high degree of conservation of T130 in apicomplexan MyoAs (Tg = *T. gondii*, Pf = *Plasmodium falciparum*, Cp = *Cryptosporidium parvum*) and MyoIc, Ie and II from *Homo sapiens* (Hs) and *Dictyostelium discoideum* (Dd). **(D)** Structure of the TgMyoA (PDB ID [6DUE]) motor domain, showing the position of T130 (red) relative to the converter, U50 and L50 subdomains. The measurements underlying the data plotted in panels A and B can be found in S11 and S12 Data. CI, confidence interval; HFF, human foreskin fibroblast.

mutation on motor function are not due to large structural changes in the protein caused by the amino acid substitution.

Including KNX-002 in the in vitro motility assay with wild-type motor inhibits both the fraction of actin filaments moving and the speed of those filaments that do move, each in a dose-dependent manner (Fig 6A, top and middle panels). Combining these 2 effects into a single "filament motility index" (i.e., the fraction of filaments moving multiplied by the mean speed of the filaments that do move) provides a better measure of the overall effect of the compound in the assay. KNX-002 induces a large decrease in the filament motility index (Fig 6A, bottom panel).

In contrast to the wild-type motor, KNX-002 had little to no effect on the fraction of actin filaments moved by the motor containing the T130A mutation (Fig 6B, top panel). Furthermore, although the basal filament sliding speed of the mutant motor was lower than that of wild type, this reduced sliding speed was also insensitive to compound (Fig 6B, middle panel), as was the mutant motor's overall filament motility index (Fig 6B, bottom panel).

Taken together, these data confirm that the T130A mutation renders TgMyoA less sensitive to the inhibitory effects of KNX-002.

## Validation of the specificity of KNX-002 in parasites

The resistant parasite line generated by mutagenesis and selection (clone R3, Fig 5B) is expected to contain as many as 70 different mutations throughout its genome [40] in addition to the T130A mutation. To confirm that resistance to KNX-002 in this parasite line was due to the identified mutation in TgMyoA, we recreated the single T130A mutation in the wild-type background using CRISPR/Cas9 (S10 Fig). To quantitatively evaluate effects of the mutation on the parasite's lytic cycle, we compared wild-type and T130A parasites in a plaque assay,

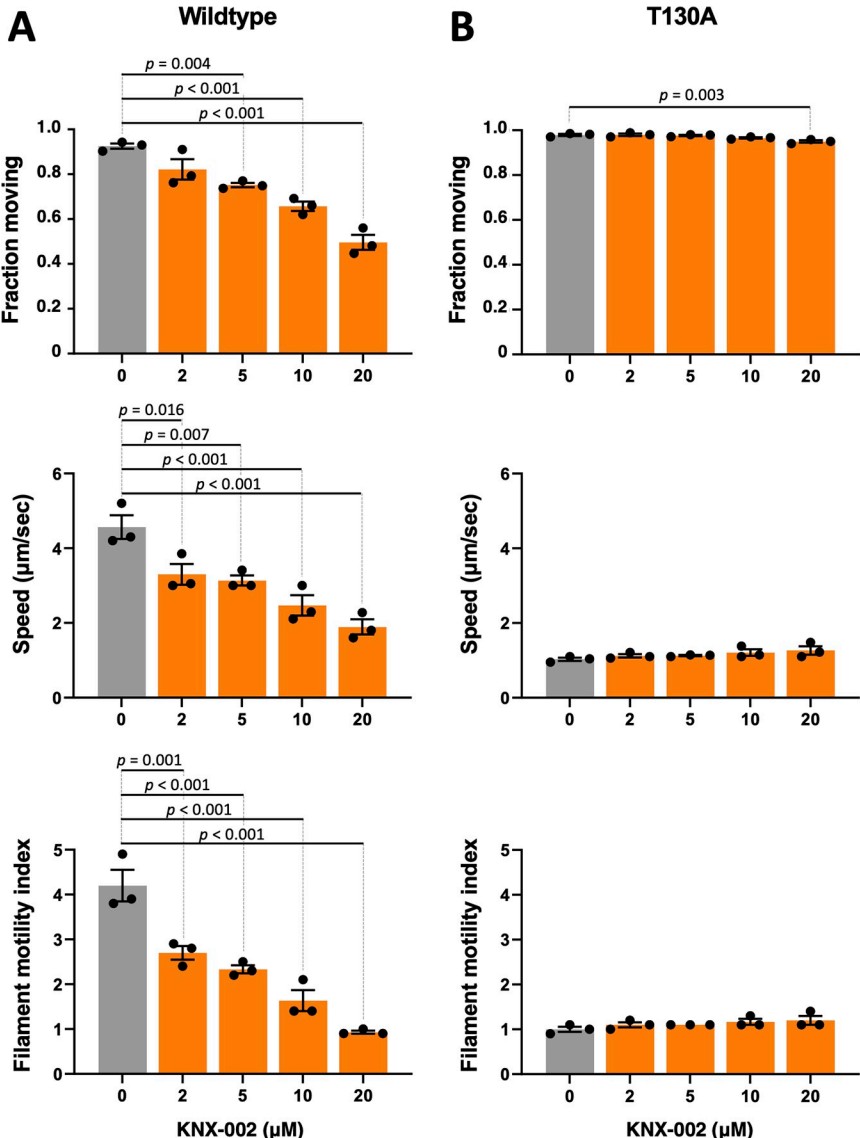

**Fig 6. KNX-002 inhibits in vitro motility driven by wild type but not T130A TgMyoA.** In vitro motility assays, showing the fraction of actin filaments moving, their sliding speed, and the corresponding filament motility index (fraction moving × mean speed) of TgMyoA treated with various concentrations of KNX-002, as indicated (0 = DMSO vehicle only). **(A)** Wild-type TgMyoA, **(B)** T130A TgMyoA. Bars show the mean of 3 independent experiments ± SEM. For each motility parameter, compound treatments were compared to DMSO by one-way ANOVA with Dunnett's test for multiple comparisons. Only the statistically significant differences ($p < 0.05$) are indicated. The measurements underlying the data plotted in this figure can be found in S13 Data.

which encompasses the parasite's entire lytic cycle. The T130A mutation had no significant effect on the number of plaques formed or plaque area (S11A and S11B Fig), demonstrating little to no effect of the mutation on the parasite's lytic cycle or growth in culture. Consistent with the plaque assay results, the T130A mutation also had little to no effect on the fraction of the parasite population moving in the 3D motility assay (compare the leftmost pair of panels in Fig 7A and 2 sets of gray bars in Fig 7B) and only a minor effect on their mean trajectory speeds (compare the 2 sets of gray bars, Fig 7C). Finally, the T130A mutation had no detectable effect on the virulence of the parasite in a mouse model of infection (S11C Fig).

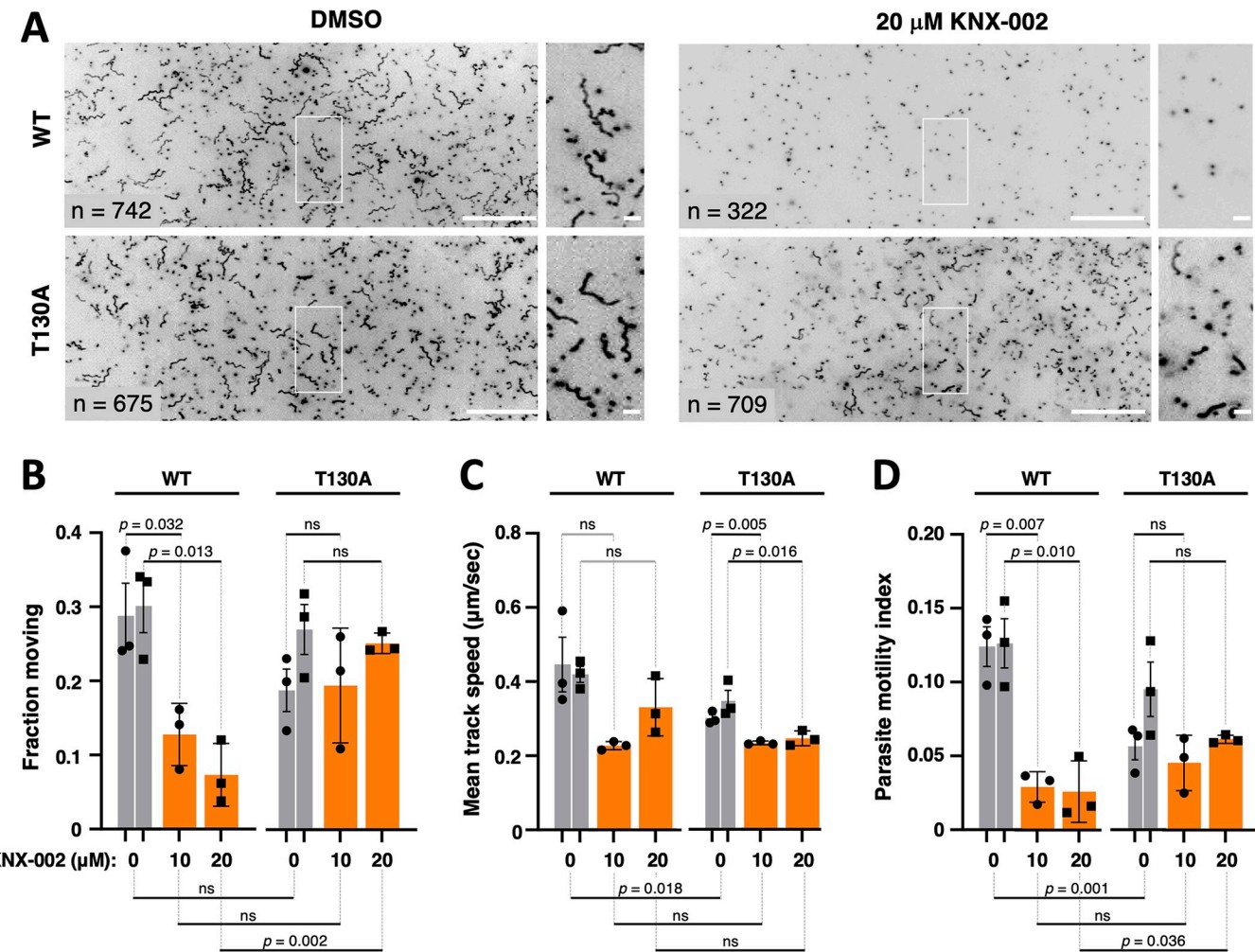

**Fig 7. The TgMyoA T130A mutation reduces parasite sensitivity to KNX-002 in 3D motility assays. (A)** Large panels: representative maximum intensity projections showing the trajectories of wild-type (WT) and T130A mutant parasites during 80 s of motility in Matrigel, in the presence of DMSO (vehicle control) or 20 µM KNX-002. The grayscale images were inverted to provide clearer visualization of the trajectories; scale bar = 100 µm. The boxed area in each large panel is magnified and shown to the right; scale bar = 10 µm. **(B)** Percent of WT and T130A mutant parasites moving in the presence of DMSO (vehicle), 10 µM KNX-002, or 20 µM KNX-002. **(C)** The mean speeds of the parasites analyzed in panel B. **(D)** The corresponding parasite motility index (fraction of parasites moving × mean speed). Each data point in panels B–D represents a single biological replicate composed of 3 technical replicates. Three of the biological replicates were collected on the same 3 days (circles), and 3 of the biological replicates were collected on a different 3 days (squares). Bars show the mean of the biological replicates ± SEM. Sets of biological replicates using the same parasite line and collected on the same days were compared by Student's one-tailed paired *t* tests (significance indicated above the graphs). Sets of biological replicates comparing different parasite lines were analyzed by Student's two-tailed unpaired *t* tests (significance indicated below the graphs). ns = not significant ($p > 0.05$). The measurements underlying the data plotted in Panels B–D can be found in S18 and S19 Data. WT, wild-type.

We next asked whether parasites expressing the T130A mutation were less sensitive to KNX-002 in the 3D motility assay. As shown above (Fig 4C), KNX-002 strongly inhibits the fraction of wild-type parasites that move in Matrigel (Fig 7B, left 3 sets of bars). In contrast, the fraction of T130A parasites that move during the assay was insensitive to KNX-002, up to concentrations of 20 µM (Fig 7B, right 3 sets of bars). The parasite speed data are more variable (Figs 7C and S7B) but show that the mutant parasites retain some sensitivity to the compound (Fig 7C, right 3 sets of bars). To capture the combined impact of changes to these 2 motility parameters and to facilitate comparison to the in vitro motility results with recombinant motor, we calculated the "parasite motility index" (the fraction of parasites moving multiplied

by the average speed of those that do move) under each condition. The wild-type parasite motility index is more severely impacted by KNX-002 treatment than the motility index of T130A parasites (Fig 7D).

The displacement of both wild-type and mutant parasites is also inhibited by KNX-002 (S7A and S12 Figs), but the mutants are again less sensitive to the compound, i.e., despite moving shorter distances in the absence of compound (compare the 2 sets of gray bars in S12 Fig), the T130A parasites moved farther than wild type in the presence of 20 μM KNX-002 (compare the third and sixth bars in S12 Fig). The decreased sensitivity of the T130A parasites to the effects of KNX-002 is apparent in the maximum intensity projections from the 3D motility assay, which show more T130A parasites than wild type moving in the presence of 20 μM KNX-002, with longer average displacements (compare right 2 sets of panels in Fig 7A).

Plaque assays were used to compare the growth of wild-type and mutant parasites in culture in the presence of KNX-002. Parasites expressing the mutant motor formed a significantly larger number of plaques in the presence of 20 μM compound than wild-type parasites, and even at 40 μM KNX-002 the mutant parasites were still able to generate small plaques (Fig 8).

These data demonstrate that the T130A mutation confers partial resistance to the compound's effects on motor function, parasite motility, and parasite growth, confirming that TgMyoA is a biologically relevant target for KNX-002 in *T. gondii*.

## KNX-002 treatment slows disease progression in mice infected with a lethal dose of *T. gondii*, through a specific effect on TgMyoA

Finally, we tested whether KNX-002 (20 mg/kg, administered intraperitoneally on the day of infection and 2 days later) can alter the progression of disease in a mouse model of infection. Only 20% of the mice that were infected with wild-type parasites and treated with vehicle were still alive 8 days postinfection, and none survived to day 9 (Fig 9, top panel, black circles). In contrast, 90% of the mice treated with KNX-002 were still alive on day 8, 80% survived to day 9, and 40% were still alive on day 11 when the experiment was terminated (Fig 9, top panel, orange squares).

When the same experiment was done with parasites expressing the T130A mutant myosin, the vehicle-treated mice did not survive beyond day 8 (Fig 9, bottom panel, black circles), similar to what we observed with wild-type parasites. In contrast to infection with wild-type parasites, KNX-002 treatment did not significantly improve the survival of mice infected with the T130A parasites (Fig 9, bottom panel, orange squares).

Together, these data show that KNX-002 can slow disease progression in mice infected with *T. gondii* and demonstrate that the compound exerts its effects through inhibition of its intended target, TgMyoA.

## Discussion

The intense interest in motility among those who study apicomplexan parasites reflects both the unique nature of the process and its importance in parasite biology and virulence. These parasites move without cilia, flagella, or a protruding leading edge that drives the substrate-dependent motility of most other eukaryotic cells. The class XIV myosins play a central role in motility, but they are unusual myosins in many respects [17,23], and precisely how these myosins and their interacting proteins function to drive motility remains controversial [27,42–47]. The mechanisms underlying motility are therefore of fundamental cell biological interest. Since MyoA is essential for virulence, highly conserved in apicomplexan parasites, and different in several respects from the myosins found in humans, the MyoA motor complex also represents a potentially attractive target for drug development. We describe here the identification

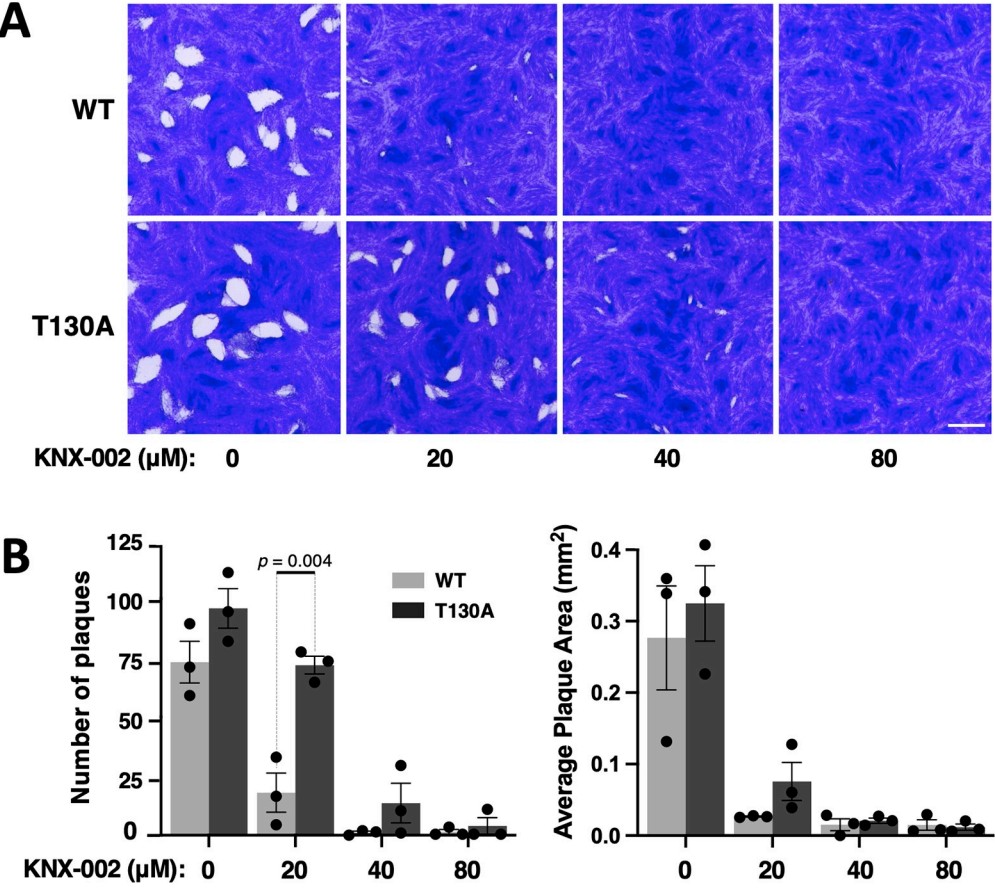

**Fig 8. The TgMyoA T130A mutation reduces parasite sensitivity to KNX-002 in growth assays. (A)** Representative images of plaque assays using wild-type (WT; top row) and T130A mutant (bottom row) parasites in the presence of 0 (DMSO vehicle only), 20, 40, and 80 μM KNX-002, 7 days after inoculating confluent HFF monolayers with 200 parasites/well. An $11.3 \times 11.3$ mm area from the middle of the well is shown; scale bar = 2 mm. **(B)** Total number of plaques per well and average plaque area of HFF monolayers inoculated with 200 parasites/well (WT or T130A) and grown for 7 days in the presence of 0 (DMSO), 20, 40, or 80 μM KNX-002. Bars show the mean of 3 biological replicates ± SEM. Treatments were compared by Student's two-tailed unpaired *t* test; only the statistically significant differences ($p < 0.05$) are indicated. The measurements underlying the data plotted in panel B can be found in S20 Data. HFF, human foreskin fibroblast; WT, wild-type.

and characterization of a novel small molecule, KNX-002, which will be a useful chemical probe for future studies of the function of TgMyoA. Importantly, we also identified a mutation in TgMyoA that reduces the sensitivity of the motor to KNX-002. We used isogenic parasite lines expressing either the wild-type or mutant myosin to demonstrate that TgMyoA is a biologically relevant target of the compound in parasites and to show that pharmacological inhibition of TgMyoA can slow the progression of disease in mice infected with a lethal dose of *T. gondii*.

## KNX-002 as a chemical probe for studying class XIV motor function

Most previous studies of TgMyoA motor function involved phenotypic characterization of parasites in which the genes encoding TgMyoA or its associated proteins were either disrupted or their expression down-regulated (e.g., [26,27,46,48]). Such reverse genetic approaches, particularly gene disruptions, are powerful but blunt tools for studying the mechanisms underlying processes important to parasite survival, such as motility. Pharmacological inhibitors that

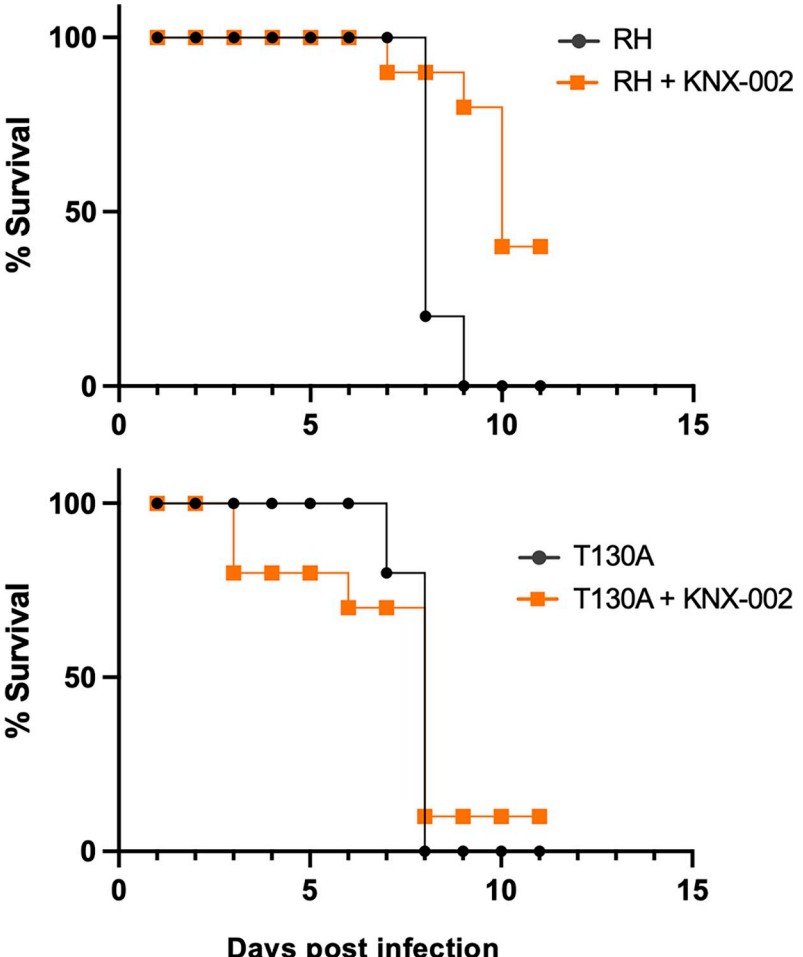

**Fig 9. KNX-002 slows disease progression in mice infected with wild-type but not T130A mutant parasites.** Four groups of mice were injected intraperitoneally on day 0 with either KNX-002 (20 mg/kg) or the equivalent volume of DMSO (vehicle). Thirty minutes later, the mice were infected with 500 wild-type parasites (RH; top panel) or 500 parasites expressing T130A mutant myosin (bottom panel). On day 2, the mice were treated again either with compound or DMSO. Differences between the survival curves were analyzed using a log-rank Mantel–Cox test ($p = 0.004$ for RH vs. RH + KNX-002 [top panel]; $p = 0.922$ for T130A vs. T130A + KNX-002 [bottom panel]) and $p = 0.008$ for RH + KNX-002 vs. T130A + KNX-002 (orange curves). The measurements underlying the data plotted in this figure can be found in S21 Data.

interact specifically with a target of interest provide a useful complementary approach for studying motility and other facets of parasite biology [49,50], since different aspects of the target's function can be perturbed at specific times and in a controlled manner. Because the parasites are exposed to the compound for only short periods of time, compensatory adaptations (e.g., through mutation or changes in gene expression) that can confound data interpretation in experiments involving gene disruption [31,32] are unlikely to occur. Even experiments involving controlled protein depletion or transcriptional down-regulation of a gene of interest, which can be achieved in hours to days, run the risk of inducing compensatory changes in transcription, translation, protein stability, posttranslational modifications, or the localization of proteins with overlapping function (e.g., [30–32]). Determining the effect of a pharmacological inhibitor of motility can, in contrast, be accomplished within minutes (e.g., Figs 4 and 7).

There have been only 2 previously reported attempts to pharmacologically manipulate the function of TgMyoA. In the first, the general myosin inhibitor 2,3-butanedione monoxime

(BDM) was shown to inhibit parasite motility [38]; however, BDM is now recognized as an inhibitor with so many off-target effects that little can be concluded from its use in cells [51,52]. The other study used computational methods and a homology model of TgMLC1 bound to TgMyoA to identify 2 predicted inhibitors of TgMLC1-TgMyoA interaction [53]. These 2 compounds (C3-20 and C3-21) inhibit *T. gondii* growth with sub-micromolar IC$_{50}$s [53] but whether this effect on parasite growth is due to inhibition of TgMyoA has not been established.

The compound we have identified and characterized here, KNX-002, inhibits the ability of recombinant TgMyoA to propel actin filaments in the in vitro motility assay, and decreases the speed of the filaments that move (Fig 6A). Similar effects are seen in the 3D motility assay: compound treatment reduces both the number of parasites moving and, to a lesser extent, the speeds with which these parasites move (Figs 7B and 7C and S7B). To provide a better picture of the combined impact of changes to these 2 motility parameters, we multiplied the fraction of actin filaments or parasites moving by the mean actin filament/parasite speed to generate the filament/parasite motility index. In the in vitro motility assay with wild-type TgMyoA, fraction moving and speed contributed approximately equally to the filament motility index in both the presence and absence of the compound. In wild-type parasites, the compound-induced changes to fraction moving are larger than to speed and therefore the predominant contributor to the changes in parasite motility index upon compound treatment. Precisely how the effects of the compound on motor function in the in vitro motility assay relate to the observed changes in parasite motility will require further study. Nevertheless, the data presented demonstrate the value of KNX-002 as a new chemical probe for studying the mechanism(s) underlying the unusual form of eukaryotic cell motility used by *T. gondii*.

Excitingly, KNX-002 and a recently discovered analog named KNX-115 were shown to also inhibit PfMyoA, the TgMyoA homolog of the malaria parasite *P. falciparum* [54,55], and to block the growth of blood stage *P. falciparum* in culture and the invasion of hepatocytes by liver stage parasites [54,55]. Structure-function studies revealed that the compound sequesters PfMyoA in a state of low affinity for actin [54], consistent with a decrease in duty ratio we observe in wild-type TgMyoA treated with KNX-002 (S13 Fig).

## Effects of the T130A mutation

The T130A mutation that confers partial resistance to KNX-002 was identified by chemical mutagenesis and selection in KNX-002. While the mutagenesis/selection approach is commonly used to generate resistant parasites in *T. gondii* and other apicomplexan parasites (e.g., [56–58]), the next step after isolating resistant clones is typically to do whole-genome sequencing on the most promising of the resistant clones to identify potentially relevant mutations. We took a more direct and scalable approach in this study, by sequencing only the motor protein loci (TgMyoA, TgMLC1, TgELC1, TgELC2) in the resistant parasites, since (a) the goal of this study was to identify a direct inhibitor of motor function; and (b) we had already shown that KNX-002 can inhibit the activity of recombinant motor consisting only of TgMyoA, TgMLC1, and TgELC1 (note: since TgELC1 and TgELC2 are functionally redundant [19], a mutation in the gene encoding either could confer compound resistance in parasites). Of the 5 resistant clones whose IC$_{50}$ for KNX-002 shifted 2.5-fold or more relative to wild type in the growth assay, only one had a mutation in the coding sequence of one of the motor protein genes, T130A in TgMyoA. The source of resistance in the other 4 clones is unknown.

T130, which lies in the first β-strand at the extreme N-terminus of the transducer domain [24], is not predicted to be directly involved in the binding of KNX-002 [54], even though other aspects of the transducer may be involved [54]. Thus, the mechanism by which the

T130A mutation confers partial resistance to the compound is likely to be indirect. For example, the T130A mutation could induce long range structural changes that alter the KNX-002 binding pocket, thereby mitigating the effect of the compound. However, myosin motors depend on a complex and highly interconnected network of allosteric interactions to perform their function [59], making it difficult to predict precisely how the T130A mutation reduces the sensitivity of TgMyoA to KNX-002.

While the T130A mutation reduces the sensitivity of the motor to KNX-002, the mutation is not without its effects on motor performance, causing a 75% to 80% reduction in basal filament sliding speed in the in vitro motility assay (Fig 6B). However, when expressed in parasites the mutation had a much less pronounced effect on parasite motility (compare the first and fourth sets of gray bars in Fig 7B–7D). We explored several possible explanations for this differential effect of the T130A mutation on the activity of the recombinant motor versus motor-driven motility of the parasites. First, the mutation might alter the folding or stability—and therefore the activity—of the recombinant TgMyoA used in the in vitro assays. This seems unlikely, since the mutant motor was indistinguishable from the wild-type motor by SEC and DSF (S9 Fig). Second, the partially functional motor might be overexpressed in the mutant parasites. This possibility was ruled out by quantitative western blotting (S14A Fig). Third, the parasite might compensate for the loss of TgMyoA activity in the T130A mutants through changes in the transcription of other myosins, myosin light chains, glideosome components, actin or actin regulatory proteins. No such changes to known motility-associated proteins were observed by RNAseq analysis (S14B Fig, right panel). Analysis of all differentially regulated transcripts revealed that 57% of the genes up/down-regulated more than log2-fold were annotated as hypothetical proteins, and the remainder were largely associated with trafficking and metabolism (S14B Fig, left panel) so cannot readily explain a compensation mechanism in parasites with impaired motor function, although further studies would be required to definitively rule out this possibility. The compound might have less of an effect on motor function in the parasite due to some protein or factor present in the parasite (but not the insect cell system used to produce recombinant motor) that can counteract the deleterious effects of the mutation. Alternatively, the density of TgMyoA motors might be higher in the parasite pellicle than on the glass coverslips used in the in vitro motility assays, and therefore require higher concentrations of compound to achieve equivalent levels of inhibition. Regardless of the reason for this differential impact of the mutation on in vitro and parasite motility, it is notable that the fraction of actin filaments moving in vitro and the fraction of parasites moving within Matrigel were better correlated than the speeds of actin filament and parasite movement. The fraction of filaments moving in the in vitro motility assay may therefore be more relevant to motor-driven parasite motility than filament sliding speed.

The mutagenesis protocol used to generate the T130A mutation kills 70% of the initial parasite population and results in an average of 63 mutations per parasite across the genome [40]. It is unclear how these harsh experimental mutagenesis conditions relate to what occurs during a natural infection and therefore how readily parasites showing reduced sensitivity to KNX-002 (or, more likely, an optimized compound based on the KNX-002 scaffold—see below) would arise during treatment of a human infection. From the drug development perspective, it would be useful to know what other resistance mechanisms are available to the parasite for this class of compounds. Further characterization of the 4 partially resistant mutants identified here that have no mutations in the TgMyoA motor (S8 Fig) would be a good place to start. TgMyoA knockout parasites are still capable of a small degree of 3D motility (13% of wild-type levels [46] under the same assay conditions used here). Although controversial [46], it has been proposed that TgMyoC can to some extent functionally compensate for the loss of TgMyoA [30,31], so mutations in TgMyoC would be of particular interest. Because humans

are a dead-end host for *Toxoplasma*, even if resistance were to emerge within a treated individual through mutations in TgMyoA or some other locus in the genome, the chances of this resistance spreading human-to-human or into the zoonotic reservoir are exceedingly small.

## MyoA as a druggable target

Small-molecule inhibitors of mammalian myosins have shown potential as therapeutics (reviewed in [51]). The most well-characterized myosin inhibitor is blebbistatin, a noncompetitive inhibitor of myosin II [60]. Blebbistatin has been a useful chemical probe for studying myosin II function [51], and derivatives are being considered for cancer chemotherapy [61,62] and drug addiction [63]. A more recently discovered inhibitor, MYK-461 (mavacamten), decreases the duty ratio of cardiac muscle myosin through an effect on $P_i$ release, reduces the development of ventricular hypertrophy in a mouse model of hypertrophic cardiomyopathy (HCM; [64]), and has been approved by the US Food and Drug Administration for use in the treatment of hypertrophic cardiomyopathy patients.

The MyoA motor complex of apicomplexan parasites has long been considered an attractive target for drug development [25,33,53,65–67]. However, no previous studies have tested whether small-molecule inhibitors of TgMyoA or other apicomplexan myosins can alter the course of infection in vivo. The argument against MyoA as a drug target has been that the parasite may have alternative and/or compensatory mechanisms to enable motility in the absence of MyoA [26,27,43,45,46], meaning that simultaneous inhibition of more than one motility pathway may be necessary to provide sufficient protection against infection. However, *T. gondii* engineered to express low levels of TgMyoA are completely avirulent [28], arguing that sufficiently strong pharmacological inhibition of TgMyoA is likely, on its own, to be therapeutically useful.

Given this rationale, a major motivation behind the work reported here was to develop the tools necessary to rigorously test whether MyoA can be targeted by small molecules to alter the course of an infection. We demonstrated that it is possible to identify small molecules that inhibit parasite TgMyoA, have no detectable effect on vertebrate myosins, and show no toxicity to mammalian cells. Furthermore, one of these compounds, KNX-002, was shown to be capable of inhibiting myosin function in the parasite, thereby disrupting parasite motility and the lytic cycle. The specificity of KNX-002 for TgMyoA in these experiments was demonstrated using parasites expressing the T130A mutation, and the observation that treatment with KNX-002 decreases the susceptibility of mice to a lethal infection with wild-type, but not T130A parasites, proves that TgMyoA is indeed a druggable target, in vivo. Whether KNX-002 can also inhibit the activity of any of the parasite's other 10 myosins [68] and disrupt their biological function(s) will be the subject of future study.

While KNX-002 provided the means to rigorously test the druggability of TgMyoA, it caused weight loss and histological evidence of liver damage in the treated infected mice. Before further animal work, it will therefore be necessary to develop more potent and less toxic analogs that retain specificity for parasite myosin. Our preliminary SAR data (summarized in Fig 3) are consistent with a key role for the pyrazole, cyclopropyl, and *p*-OMe aryl units within KNX-002, with a generally good correlation between the results of the recombinant myosin ATPase and parasite growth inhibition assays. Subsequent studies, inspired both by our SAR data and the recently reported structure of the PfMyoA-KNX-002 complex [54], could explore the effect on biological activity of: (a) alternative replacements for the thiophene group; and (b) a switch in the positions of the NH and $CH_2$ groups, among other options. Encouragingly, Trivedi and colleagues recently identified an analog of KNX-002 that shows >20-fold improved potency against recombinant PfMyoA [55]. Although neither the structure of this

analog nor the SAR strategy that led to its discovery was disclosed, this result suggests there is considerable opportunity to optimize the biological properties of the KNX-002 scaffold through a systematic SAR campaign.

KNX-002 and any improved analogs developed through further rounds of iterative SAR will be useful new experimental tools for studying the role of MyoA in parasite growth, dissemination and virulence, and provide an exciting opportunity to develop an entirely new class of drugs that can be used to prevent or treat the severe diseases caused by *T. gondii* and other apicomplexan parasites.

## Methods

### Cell and parasite culture

Parasites were propagated by serial passage in HFFs. HFFs were grown to confluence in Dulbecco's Modified Eagle's Medium (DMEM) (Life Technologies, Carlsbad, California, United States of America) containing 10% v/v heat-inactivated fetal bovine serum (FBS) (Life Technologies, Carlsbad, California, USA), 10 mM HEPES (pH 7), and 100 units/ml penicillin and 100 μg/ml streptomycin, as previously described [69]. Prior to infection with *T. gondii*, the medium was changed to DMEM supplemented with 10 mM HEPES (pH 7), 100 units/ml penicillin and 100 μg/ml streptomycin, and 1% v/v FBS.

### Compounds

KNX-002 was either obtained commercially from Asinex (Winston-Salem, North Carolina, USA; Figs 1, 2, 4–8 and S1, S2, S6–S8, S11–S13) or synthesized, purified, and characterized as described below (Figs 3, 9 and S3–S5, S10). The commercially sourced and in-house synthesized compounds had similar $IC_{50}$s in parasite growth assays: see $IC_{50}$ values and overlapping 95% CIs in Figs 2 and 5 (commercial) and Fig 3 (synthesized). Compounds were dissolved in dimethyl sulfoxide (DMSO) and stored at −20°C, in the dark, until use. In all experiments involving treatment with compound, an equivalent amount of DMSO was added to all samples in that experiment, including the DMSO only vehicle controls. Final concentrations of DMSO were 0.5% v/v for experiments with recombinant TgMyoA, 0.1% v/v for experiments with parasites and cells, and 34 μl DMSO per injection per 20 g mouse for the in vivo experiments.

### High-throughput screen

Compound CK2140597 was originally identified through high-throughput screening of a library of 50,000 compounds (40 μM final concentration, 2% v/v DMSO), performed by JS and EW at Cytokinetics, to identify inhibitors of the actin-activated ATPase of TgMyoA, provided by AK and GEW. The same compounds were screened, in parallel, against *Plasmodium falciparum* MyoA (PfMyoA; protein provided by Kathy Trybus, University of Vermont). The assay used was a coupled enzymatic assay [70]. Actin was first polymerized using 1 mM ATP, 2 mM $MgCl_2$, 20 mM KCl and left on ice for 30 min. Buffer A was made with PM12 (12 mM Pipes-KOH (pH 7.0), 2 mM $MgCl_2$), 1 mM DTT, 0.1 mg/ml BSA, 0.4 mM PK/LDH, 1 μg/ml TgMyoA, 0.009% v/v Anitfoam (Millipore Sigma, St. Louis, Missouri, USA). Buffer B was made with PM12, 1 mM DTT, 0.1 mg/ml BSA, 1 mM ATP, 1 mM NADH, 1.5 mM PEP, 0.6 mg/ml previously polymerized actin and 0.009% v/v Antifoam. Approximately 12.5 μl of each buffer were added to a plate with a Packard MiniTrak liquid handler and then mixed at 2,400 for 45 s with a BioShake 3000 ELM orbital plate mixer (Qinstruments). The decrease in optical density at 340 nm as a function of time was measured on an Envision 2104 spectrophotometer (Perkin Elmer) with a 600 s data monitoring window sampled every 60 s. Hit progression

included confirmation of activity in an independent experiment, inactivity against an unrelated ATPase (hexokinase from Millipore Sigma), and dose-responsive inhibition. Hexokinase reactions were performed using glucose (1 mM) and sufficient hexokinase to generate ADP from ATP at a rate similar to the target myosins. Based on preestablished thresholds, CK2140597 was identified as a hit in the TgMyoA but not the PfMyoA screen; however, resupplied compound was subsequently shown to also be active against PfMyoA by Trybus and colleagues. CK2140597 was licensed by Kainomyx (www.kainomyx.com), renamed KNX-002, and released for academic studies in the laboratory of GEW (for *Toxoplasma*) and Kathy Trybus (for *Plasmodium*).

## Vertebrate myosin ATPase assays

Myofibril preparations from striated muscles are a well-established model for assaying myosin ATPase activity in a semi-native context [71]. Bovine cardiac, rabbit fast skeletal, and bovine masseter slow skeletal myofibrils were prepared and assayed as previously described [72], except they were tested at 75% calcium activation (pCa75) since this high level of basal ATPase activity enables the sensitive detection of inhibitors. For SMM, native chicken gizzard SMM S1 was prepared, crosslinked to bovine cardiac actin using EDC/NHS, and assayed as previously described [73].

## Expression and purification of recombinant TgMyoA

TgMyoA was co-expressed with TgMLC1, TgELC1, and TgUNC in Sf9 cells and purified as previously described [33]. TgMyoA containing the T130A mutation was generated using the QuikChange II-Site-Directed Mutagenesis method (Agilent Scientific, Santa Clara, California, USA). Briefly, mutant strand synthesis was performed by denaturing baculovirus plasmid pSmTgMyoAcBiocfFLAG, annealing primers P21 and P22 (S1 Table), which contain the desired mutation and using PfuUltra high fidelity taq polymerase in a thermocycling reaction to incorporate and extend the primer product. The product was then digested by DpnI overnight at 37°C and the resulting plasmid was purified using a Qiagen PCR purification kit. Subsequently, the purified PCR product was used to transform electro-competent DH5-alpha cells. Positive clones were screened, plasmid prepped and the presence of the mutation verified by sequencing.

## In vitro motility

In vitro motility assays with chicken skeletal muscle actin were performed as previously described [33] with the following modifications: (a) Neutravidin (40 µg/ml, Molecular Probes) in buffer B was added to the flow cells first, followed by BSA (in buffer B) and 3 washes with buffer B; (b) approximately 0.16 µg of FLAG purified TgMyoA was used per flow cell; (c) buffer C was in all cases supplemented with oxygen scavengers (3 mg/ml glucose (Sigma-Aldrich), 0.125 mg/ml glucose oxidase (Sigma-Aldrich), and 0.05 mg/ml catalase (Sigma-Aldrich)); (d) all washes with buffer C were done 3 times; and (e) bacterially expressed TgMLC1 and TgELC1 were each added to the flow cells at a concentration of 25 µg/ml. Data were collected on an Eclipse Ti-U inverted microscope (Nikon) equipped with a 100× Plan Apo objective lens (1.49 NA) and a XR/Turbo-Z camera (Stanford Photonics) running Piper Control software (v2.3.39). Data were collected at 10 frames per second. Actin filament tracking and analysis was done in Imaris ×64 v. 7.6.1 (Bitplane AG, Zurich, Switzerland).

## Duty ratio measurements

To compare the duty cycle of WT recombinant TgMyoA with and without KNX-002 treatment, the number of myosin heads capable of interacting with an actin filament was estimated

as described previously [20,74]. Briefly, the in vitro motility assay was carried out as described above using various concentrations of myosin treated with either 10 μM KNX-002 or the equivalent amount (0.5% v/v) of DMSO. At each concentrations of myosin on the coverslip, actin filament length and velocity was measured in Imaris ×64 v. 7.6.1 (Bitplane AG, Zurich, Switzerland) and an estimate of the number of heads available to interact with an actin filament was determined. Duty cycle (f) were calculated by plotting filament velocity (V) against number of heads (N). These data were then fit to the equation $V = (a \times V_{max}) \times [1-(1-f)^n]$ by nonlinear regression analysis (GraphPad Prism (Version 9.4.0 (453))) as previously described [74,75].

## Parasite growth assays

**Fluorescence-based growth assay.** To quantify growth in the presence of compound(s), $1 \times 10^4$ tachyzoites were preincubated with compound for 5 min prior to inoculation onto 3 wells of a black sided, clear bottom 384-well plate containing a confluent monolayer of HFF cells. For compound assays, parasites expressing tdTomato were incubated with vehicle or serial dilutions of compound, maintaining a constant DMSO concentration (0.2% v/v). The fluorescent signal was quantified daily for 7 days (ex 530/25, em 590/35) on a Biotek synergy 2 plate reader. The $IC_{50}$ of growth inhibitory compounds was quantified using GraphPad Prism.

**Plaque assay.** 10, 100, 200, or 500 parasites were used to inoculate a confluent monolayer of HFF cells in a 12-well plate and incubated for 7 days at 37°C, 5% $CO_2$. Infected monolayers were then washed with phosphate-buffered saline, fixed with 100% EtOH for 5 min, stained with 5× crystal violet solution (12.5 g crystal violet, 125 ml 100% EtOH, 500 ml 1% w/v ammonium oxalate) for 5 min, washed with PBS, and allowed to dry. Wells were imaged using Bio-Tek ImmunoSpot S6 MACRO Analyzer, and plaque number and area were measured using the Analyze Particles tools in Fiji v2.3.0.

## Parasite motility assays

2D motility assays were performed as previously described [76], using monoclonal antibody DG52 (generously provided by Dr. David Sibley) against TgSAG1 for immunofluorescent staining of the trails.

3D motility assays were performed as previously described [16]. Briefly, parasites were syringed released and filtered using a 3-μM Whatman Nuclepore filter (Millipore Sigma) before spinning and resuspending in motility buffer containing 0.5 mg/ml Hoechst 33342. A mixture of 1:3:3 of parasites, motility buffer with compound, and Matrigel was created and pipetted into a flow cell. Flow cells were incubated for 7 min at 27°C and then 3 min at 35°C. Images were captured using a Nikon Eclipse TE300 epifluorescence microscope (Nikon Instruments, Melville, New York, USA) using a 20× PlanApo λ objective and a NanoScanZ piezo Z stage insert (Prior Scientific, Rockland, Massachusetts, USA). For the data shown in Figs 4B and 4C and S7, an iXON 885 EMCCD camera (Andor Technology, Belfast, Ireland) driven by NIS Elements software v. 3.20 (Nikon Instruments, Melville, New York, USA) was used to collect time lapse image (502 pixel × 500 pixel) stacks of the fluorescently label parasite nuclei, with Mercury lamp excitation. Images were captured for 60 s. The camera was set to trigger mode, 2 × 2 binning, readout speed of 35 MHz, conversion gain of 3.8×, and EM gain setting of 3. For the data shown in Figs 7 and S12, 1,024 pixel × 384 pixel images were captured using an iXON Life 888 EMCCD camera (Andor Technology) driven by NIS Elements software v.5.11, including the Illumination Sequence acquisition module (Nikon Instruments), and excited using a pE-4000 LED illumination system (CoolLED, Andover England). Images were captured for 80 s. The camera was set to trigger mode, no binning, readout speed of 30 MHz, conversion gain of 3.8×, and an EM gain of 300.

All 3D motility assays captured image stacks consisting of 41 z-slices, captured 1 μm apart with a 16 ms exposure time. The data were analyzed using Imaris ×64 v. 9.2.0 (Bitplane AG, Zurich, Switzerland). The ImarisTrack module tracked the parasite nuclei using an estimated spot volume of $3.0 \times 3.0 \times 6.0$ μm. A maximum distance of 6.0 μm and maximum gap size of 2 were applied to the tracking algorithm. A track duration filter of 10 s was used to avoid tracking artifacts and a 2 μm displacement filter was used to eliminate parasites moving by Brownian motion [16].

## Mutagenesis and selection for resistant parasites

The dose of ENU required to kill 70% of parasites [40] during a 4-h treatment was determined by plaque assay (see above). For each mutagenesis, tdTomato-expressing parasites were liberated from large vacuoles by scraping the monolayer and syringe release through a 26G needle, then treated with ENU at the 70% kill concentration for 4 h. The parasites were washed 3× in DMEM containing 1% v/v FBS to remove residual ENU and inoculated onto a T25 of confluent HFF cells, and 24- to 48-h postinfection, surviving parasites that had lysed their host cells were transferred to fresh HFF cells and grown for until plaques became visible (10 to 14 days) in the presence of 40 μM KNX-002, which is close to the compound's estimated $EC_{90}$ value in the growth assay (37.6 μM; 95% CI 18.9 to 136.4 μM). Clonal lines were isolated by serial dilution in five 96-well plates per mutagenesis. Parasite sensitivity to KNX-002 was then determined by fluorescence-based growth assay (see above).

RNA was isolated from resistant clones with the GeneJET RNA purification kit and used to generate cDNA with the Agilent cDNA synthase kit for oligoDT synthesis. cDNA encoding TgMyoA, TgMLC1, TgELC1, TgELC2 was amplified and sequenced with primers covering the entire transcript (S1 Table, P6-P20). Sequence alignments were analyzed in SnapGene and sequences containing non-synonymous mutations were compared to other strains to identify naturally occurring polymorphisms unlikely to be responsible for resistance.

## Generation of TgMyoA T130A mutant using CRISPR/Cas9

All oligonucleotides were synthesized by Sigma-Aldrich (The Woodlands, Texas, USA), and restriction enzymes were purchased from New England Biolabs (Ipswich, Massachusetts, USA). To generate the Cas9eGFPTgMyoAguide plasmid (S10 Fig), plasmid pSAG1::Cas9-U6::sgUPRT [77] was PCR amplified with primers P1 and P2 (S1 Table). This linear pSAG1::Cas9-U6::sgUPRT PCR product contained all features of the original vector, the TgMyoA T130 protospacer sequence (5′-GAATCAGATCTACGCGACTG-3′), and introduced an EcoRI site. The T130A double-stranded repair template was produced by annealing oligonucleotides P3 and P4 (S1 Table). This repair template contains the T130A mutation and 2 additional mutations that do not alter the amino acid sequence: a mutation introducing a diagnostic BglII site and a mutation ablating the PAM sequence (S10A Fig). RHΔku80Δhxgprt parasites were transfected with the ds repair template and Cas9eGFPTgMyoAguide plasmid (S10B Fig). Wild-type RHΔku80Δhxgprt and transfected tachyzoites were syringe released and filtered 2 days post-transfection. eGFP+ parasites were sorted 3-per well onto a 96-well plate in the UVM Flow Cytometry and Cell Sorting Facility using a FACSAria cell sorter (Becton Dickinson Biosciences, Franklin Lakes, New Jersey, USA). Positive clonal populations were validated for GFP expression 7 days post-sorting.

To identify T130A positive mutant clones, gDNA was purified from WT and transfected parasites directly from 96-well plates using previously methods [78]. PCR fragments of approximately the first 3 kb of MyoA were generated using the Phire tissue direct PCR Master kit (Thermo Fisher Scientific, Waltham, Massachusetts, USA) with primers P5 and P11 (S1

Table). Diagnostic digests with BglII provided an initial screen for T130A mutant parasites as the ds repair template introduced a third BglII site into the amplified sequence (S10 Fig). Positive T130A mutant clones were further verified by harvesting tachyzoite cDNA and sequencing using primers P5-P13 (S1 Table).

## Western blotting

Protein samples were resolved by SDS–polyacrylamide gel electrophoresis and transferred to Immobilon-FL polyvinylidene fluoride membrane (Millipore Sigma) as previously described [76]. Blots were probed with rabbit anti-TgMyoA antiserum UVT109 (diluted 1:1,000) and rabbit anti-TgACT1 (1:10,000; generous gift from Dr. David Sibley). Goat anti-rabbit IRDye800CW (1:20,000; LI-COR Biosciences, Lincoln, Nebraska, USA) was used as a secondary antibody. The blots were scanned using an Odyssey CLx infrared imager (LI-COR) and analyzed using Image Studio software v.2.1.10 (LI-COR).

## Mammalian cell toxicity assays

HFF and HepG2 cells were grown to approximately 80% confluence in 384-well plates before media was replaced with DMEM containing 10% v/v FBS and serial dilutions of compound or vehicle while maintaining a consistent DMSO concentration (0.2% v/v). Cells were incubated with compound at 37°C for 72 h before multiplexed quantitation of toxicity using CellTox green and cell viability using CellTiter-Glo (both assay kits from Promega Corp, Madison, Wisconsin, USA). As a positive control for toxicity 24 h prior to cessation of the assay 0.5 mM NaN$_3$ was added to half of the vehicle wells to induce cell death.

**CellTox green.** To quantify dead, dying and damaged cells in each well, an equal volume (25 μl) of CellTox green reagent (1:500 CellTox green dye in CellTox green assay buffer) was added to each well and mixed on an orbital shaker (700 to 900 rpm) for 1 min. The plate was equilibrated at room temperature for 15 min before quantifying the fluorescence signal (ex 485/20, em 528/20) using a Biotek Synergy 2 plate reader.

**CellTiter-Glo.** Following quantification of toxicity, the number of viable cells/well was quantified by adding 25 μl CellTiter-Glo reagent to each well. To maximize cell lysis plates were mixed on an orbital shaker (700 to 900rpm) for 5 min before equilibration at room temperature for an additional 10 min. Quantitation was achieved by measuring luminescence per well using a Biotek Synergy 2 plate reader.

## Parasite replication assay

Confluent HFF monolayers grown in clear bottomed black-sided 96-well plates were challenged with $1 \times 10^4$ tachyzoites and incubated for 1 h at 37°C with 5% CO$_2$. Monolayers were then washed once with DMEM containing 1% v/v FBS before incubating with DMSO vehicle only or compound. Monolayers were fixed with 2% v/v PFA 12 h, 24 h, and 36 h after compound addition. After blocking and permeabilization in 0.2% v/v Triton X-100 and 1% w/v bovine serum albumin, samples were incubated with Mab 45.36 against TgIMC1 for 2 h (3.7 μg/ml) at 23°C. Monolayers were washed 3 times with blocking buffer before incubation in the dark with a 1:1,000 dilution of Alexa546-conjugated anti-rabbit IgG (Thermo Fisher) for 1 h at 23°C. The number in parasites in each of 50 vacuoles per sample were counted blind.

## Parasite toxicity assays

**Extracellular parasite toxicity.** The $1 \times 10^7$ parasites in DMEM containing 1% v/v FBS were mixed in a 96-well plate with 0 (DMSO vehicle only), 40, or 80 μM KNX-002, in a final

volume of 50 μl. The plate was incubated for 1 or 3 h at 37°C with 5% $CO_2$. Approximately 50 μl CellTiter-Glo reagent was added to each well and mixed on an orbital shaker for 5 min (500 to 700rpm), followed by a 15-min incubation at 23°C. The relative number of viable parasites was quantified by luminescence, read on a Biotek Synergy 2 plate reader.

**Intracellular parasite toxicity.** Intracellular parasite toxicity in the presence of various concentrations of compound was determined using the replication assay described above.

## ATPase assays

ATPase assays for the SAR studies were performed per manufacturer's recommendations using the ADP-Glo Kinase Assay (Promega Corp, Madison, Wisconsin, USA). Briefly, 8 μl kinase reactions were performed by combining 5 μl of myosin (0.128 mg/ml), 1 μl of actin (4.8 mg/ml), 1 μl of ATP (8 mM), and 1 μl of 1× Kinase Reaction Buffer (40 mM Tris (pH 7.6), 20 mM $MgCl_2$, and 0.1 mg/ml BSA). After a 90-min incubation at room temperature, 5 μl of ADP-Glo Reagent was added to the kinase reaction followed by a 40-min incubation at room temperature. Finally, 10 μl of Kinase Detection Reagent was added and after a 60-min incubation, ATP production was detected by measuring luminescence with a Synergy H1 Hybrid plate reader. For compound inhibitor experiments, the control reactions contained 0.5% v/v DMSO and compounds were diluted in 1× Kinase Reaction Buffer. All assays were performed under conditions where ADP generation (measured in relative luminescence units, RLUs) was linear with respect to both TgMyoA concentration and incubation time. Reactions were performed in duplicate on each plate and all experiments were replicated 3 times. Data were analyzed using GraphPad Prism.

## Size-exclusion chromatography

Twin-strep-II (TST)-tagged TgMyoA (1–778) (ToxoDB ID TGGT1_070410/GenBank accession number EPR60343.1) wild-type and T130A mutant were expressed in *Trichoplusia ni* (Tni) insect cells. Protein was purified by streptactin affinity, and finally SEC, using a final buffer containing 20 mM HEPES (pH 7.5), 150 mM NaCl, and 1 mM DTT. Prior to DSF experiments, purified, concentrated TgMyoA (1–778) was diluted with buffer containing $MgCl_2$, ADP, NaF, and $Al(NO_3)_3$, to final concentrations of 2 mM $MgCl_2$, 2 mM ADP, 6 mM NaF, 1.75 mM $Al(NO_3)_3$.

## Differential scanning fluorimetry

Purified TgMyoA (1–778) WT and T130A were diluted to 1.0 mg/mL in buffer containing 150 mM NaCl, 20 mM HEPES (pH 7.5), 1 mM $MgCl_2$, 1 mM ADP, 6 mM NaF, 1.75 mM $Al(NO_3)_3$, 1 mM DTT, and SYPRO orange dye at a final concentration of 5×. Approximately 20 μl reactions were tested in triplicate using the StepOnePlus Real-Time PCR System from Thermo Fisher, with all replicate samples obtained from a single preparation of MyoA protein.

## Mouse infections

Infections were carried out in 4- to 5-week-old male CBA/J mice purchased from Jackson Laboratories. All mice were socially housed and acclimated for 3 days prior to infections. For virulence studies, a total of 9 mice (3 biological replicates of 3 mice each) were challenged by intraperitoneal injection of either 50 or 1,000 wild-type (RH) or T130A parasites resuspended in a total volume of 200 μl sterile PBS. Daily animal health monitoring began 2 days after infection. Mice were euthanized when predetermined welfare thresholds were exceeded.

For KNX-002 treatment studies, 15 mice were injected intraperitoneally with either 20 mg/kg KNX-002 (40 mM compound stock in DMSO diluted into PBS containing 30% v/v polyethylene glycol 400) or an equivalent amount of DMSO (vehicle) in the same diluent, and 20 to 30 min after compound/vehicle injection, 10 compound-treated mice were intraperitoneally infected with either 500 wild-type or 500 T130A mutant parasites, and 5 vehicle-treated mice were similarly infected with either wild-type or T130A mutant parasites. A second intraperitoneal injection of 20 mg/kg KNX-002 or DMSO was given 2 days later. The rationale for this dosing schedule was 2-fold: (a) We expected the compound to work primarily on egressing and extracellular parasites. Since the parasite's lytic cycle takes approximately 48 h, this 2-day injection schedule was chosen to maximize exposure of the extracellular parasites to freshly injected compound early in establishment of the infection. (b) A preliminary KNX-002 dose tolerance study was conducted prior to the infection experiments. Two doses of 20 mg/kg (the highest dose tested) administered 2 days apart revealed minor hepatoxicity without signs of excessive pain or distress and was therefore considered the maximal tolerated dose.

Daily animal health monitoring began 2 days after infection, and mice were euthanized when predetermined welfare thresholds were exceeded. All animal experiments were carried out under University of Vermont Institutional Animal Care and Use Committee protocol #PROTO202000126.

### RNA sequencing

For each sample, 3 confluent flasks of HFFs cells were inoculated with either wild-type or T130A mutant parasites. Two days after infection, when vacuoles were large ($\geq$64 parasites/vacuole), parasites were harvested by scraping the flask and passing the suspension 3 times through a 26G needle. Released parasites were incubated at 37°C for 1 h, and RNA was then isolated using the GeneJET RNA Purification kit (Thermo Fisher) following manufacturer instructions. For each sample, 1 μg of total RNA (A260/280 = 1.8–2.2, A260/230 $\geq$ 1.8) was sent to Novogene for library preparation and sequencing to a read depth of 20 million paired end reads per sample. Quality control and sequence alignment to the *Toxoplasma* genome (toxodb.org) was done in house. The most differentially expressed genes were identified from all genes expressed in all samples, with a FPKM > 0.1 and an adjusted *p*-value of <0.05. Expression of known TgMyoA interacting proteins, glideosome components, actin and actin regulatory factors were also separately quantified where detected in all 3 replicates of both the wild-type and T130A mutant. TgELC2 (TGME49_305050) was not consistently expressed and therefore was excluded from the analysis shown in S14B Fig.

### Compound synthesis and characterization

Details of compound synthesis and analysis are presented in S1 Text. Mass spectroscopic (m/z) data were obtained by Mrs. Caroline Horsburgh at the University of St Andrews mass spectrometry service. Electrospray ionization techniques were applied and a Thermo Fisher LTQ Orbitrap XL spectrometer was used. Values are quoted as a ratio of mass to charge (m/z) in Daltons [Da]. NMR spectra were recorded on either a Bruker Ascend 500 ($^1$H 600; $^{13}$C 150 MHz) or a Bruker Advance II 400 ($^1$H 400; $^{13}$C 101 MHz). $^{13}$C NMR spectra were taken with a DEPTQ pulse sequence. Spectra were recorded using deuterated solvents. For $^1$H NMR analysis, multiplicities of signals are denoted as follows: s (singlet), d (doublet), t (triplet), q (quartet), dd (doublet of doublets), dt (doublet of triplets), and m (multiplet). Coupling constants (J) are quoted to the nearest 0.1 Hz. Chemical shifts (δ) are stated in ppm and referenced to residual solvent signals (CHCl$_3$ 7.260 ppm (s), 77.160 ppm (t); (CD$_3$)$_2$SO 2.50 ppm (s) or 39.52 ppm (pentet)). IR spectra were obtained using a Shimadzu IRAffinity 1S IR Spectrometer

as ATR. The IR data were processed using OriginPro 2022 v9.9.0.225. Optical rotations were determined using a Perkin Elmer Model 341 Polarimeter with a Na/Hal lamp (Na D line, 589 nm). Melting points were measured using capillary tubes on an Electrothermal 9100 melting point apparatus. Values are quoted in ranges and to 1.0°C.

## Supporting information

**S1 Fig. KNX-002 has little to no effect on the activity of various vertebrate myosins.** Dose-response curves showing the effect of KNX-002 on the actin-activated ATPase activity of various vertebrate myosins. SMM = smooth muscle myosin; pCa25 and pCa75 refer to assays done at 25% and 75% calcium activation, respectively. The measurements underlying the data plotted in this figure can be found in S1 Data.
(TIF)

**S2 Fig. KNX-002 has no effect on parasite intracellular replication.** Newly infected HFF cells were incubated at 37°C in culture medium containing 80 μM KNX-002 or an equivalent volume of DMSO (vehicle) for the indicated times. The cells were then fixed and processed for immunofluorescence using an antibody against the inner membrane complex marker IMC1, enabling the number of parasites in each of 100–120 vacuoles to be counted. Shading of the bars indicates number of parasites per vacuole (see key); the means ± SEM at each time point of 2–5 biological replicates (each consisting of 2 technical replicates) are shown. There were no significant differences between the number of DMSO- and compound-treated parasites per vacuole at each of the time points, based on unpaired Student's *t* tests using a 2-step Benjamini, Krieger, and Yekutieli correction for multiple comparisons. The measurements underlying the data plotted in this figure can be found in S3 Data.
(TIF)

**S3 Fig. SAR analysis of KNX-002 analogs: TgMyoA ATPase activity.** Actin-activated ATPase activity of wild-type TgMyoA in the presence of DMSO (vehicle), KNX-002, or each of the 15 KNX-002 analogs shown in Fig 3. All compounds were used at 20 μM. ATPase activity is expressed as relative luminescence units (RLUs). Each data point represents a single biological replicate composed of 2 technical replicates. Bars show the mean ± SEM of 2 biological replicates. RLU values were compared pairwise to KNX-002 (black asterisks) or to DMSO (red asterisks) by one-way ANOVA with Dunnett's test for multiple comparisons; asterisks indicate $p < 0.05$. The measurements underlying the data plotted in this figure can be found in S4 Data.
(TIF)

**S4 Fig. SAR analysis of KNX-002 analogs: parasite growth. (A)** tdTomato-expressing wild-type parasites were preincubated for 5 min with various concentrations (see key, bottom right) of KNX-002 or one of the 15 analogs shown in Fig 3 and then added to HFF cells on a 384-well plate. Fluorescence was measured daily over the next 7 days to quantify parasite growth. Each data point represents the mean of 3 biological replicates, each consisting of 2 to 3 technical replicates. RFU = relative fluorescence units. **(B)** The data from either day 5 (D5) or day 6 (D6) of the growth assay shown in panel A were used to calculate the $IC_{50}$ for parasite growth; ND = insufficient inhibition from 0–80 μM to determine an $IC_{50}$. The measurements underlying the data plotted in this figure can be found in S5 and S6 Data.
(TIF)

**S5 Fig. SAR analysis of KNX-002 analogs: host cell toxicity.** Subconfluent HepG2 and HFF cells were treated for 72 h with DMSO (vehicle), KNX-002, or each of the 15 KNX-002 analogs shown in Fig 3, at a final concentration of either 40 μM (light gray bars) or 80 μM (dark gray

bars). The treated cells were then subjected to CellTiter-Glo viability (top 2 panels) and Cell-Tox green cytotoxicity assays (bottom 2 panels). Colored bars on the right indicate thresholds for negligible (green), mild (yellow), and moderate (orange) loss of viability (top 2 panels) or toxicity (bottom 2 panels). The measurements underlying the data plotted in this figure can be found in S7 Data.
(TIF)

**S6 Fig. VEST5 is non-toxic to parasites. (A)** Extracellular parasite toxicity. Wild-type parasites were incubated at 37°C in culture medium containing 0 (DMSO only), 40 μM, or 80 μM VEST5 for the indicated times. Parasite viability was then determined using the CellTiter-Glo assay. Bars show the mean of 2 biological replicates ± SEM. **(B)** Intracellular parasite toxicity. Newly infected HFF cells were incubated at 37°C in culture medium containing 0 (DMSO only) or 80 μM VEST5 for the indicated times. The cells were then fixed and processed for immunofluorescence using an antibody against the inner membrane complex marker IMC1, enabling the number of parasites in each of 50 vacuoles to be counted. Shading of the bars indicates number of parasites per vacuole (see key); the means ± SEM at each time point of 2 biological replicates are shown. The measurements underlying the data plotted in this figure can be found in S8 Data.
(TIF)

**S7 Fig. Dose-dependent effects of KNX-002 on parasite motility parameters. (A)** First-to-last point trajectory displacement and **(B)** mean speed along the trajectory during a 60 s 3D motility assay in the presence of the indicated concentrations of KNX-002. Each data point represents a single biological replicate composed of 3 technical replicates. The number of trajectories included in the displacement and speed measurements decreases with increasing concentration of compound due to the reduction in percent moving (see Fig 4C). Sets of DMSO- and compound-treated parasite data captured on the same days, indicated by the similar symbol shapes, were compared by Student's one-tailed paired $t$ tests. Bars show the mean of the biological replicates ± SEM. Only the statistically significant differences ($p < 0.05$) are indicated. The measurements underlying the data plotted in this figure can be found in S9 and S10 Data.
(TIF)

**S8 Fig. Growth assays of mutant parasites showing reduced sensitivity to KNX-002. (A)** tdTomato-expressing were mutagenized, selected in 40 μM KNX-002, and cloned as described in Methods. Each of the 26 clones recovered from 2 independent rounds of mutagenesis and selection was preincubated with various concentrations of KNX-002 for 5 min and then added to HFF cells on a 384-well plate. Fluorescence was measured daily over the next 7 days to quantify parasite growth. RFU = relative fluorescence units. **(B)** $IC_{50}$ curves corresponding to the growth assay data shown in panel A, for the 5 clones that showed an $IC_{50}$ shift of >2.5-fold compared to the un-mutagenized RH (tomato) parasites. The data shown are from 3 independent biological replicates, each consisting of 2 to 3 technical replicates at all time points. Vertical bars indicate SEM in panel A and 95% CI in panel B. Clone R3 contains a T130A mutation in the gene encoding TgMyoA; none of the other 4 clones contain mutations in the genes encoding TgMyoA, TgMLC1, TgELC1, or TgELC2. The measurements underlying the data plotted in in this figure can be found in S11 and S12 Data.
(TIF)

**S9 Fig. The T130A mutation does not affect the overall structure or stability of TgMyoA. (A)** SEC UV traces of TgMyoA motor domain, wild-type (blue), and T130A mutant (red), showing identical elution profiles. **(B)** Derivative fluorescence data from DSF experiment showing minimal changes in thermostability. Values are average of triplicate data. The

measurements underlying the data plotted in this figure can be found in S14 and S15 Data. (TIF)

**S10 Fig. Generation of the TgMyoA T130A mutant parasite using CRISPR/Cas9-directed mutagenesis. (A)** Schematic of the *TgMyoA* gene (exons in dark blue, introns light blue) and the amino acids surrounding T130 (boxed) in exon 2. Mutations introduced by the double stranded (ds) repair template are denoted with asterisks (*): the leftmost mutation introduces a diagnostic BglII site, the middle mutation generates the T130A substitution (boxed), and the rightmost mutation ablates the PAM sequence. The guide sequence (protospacer) is underlined, and the PAM sequence is highlighted in gray. The location of primers P5 and P11 used to amplify a fragment from gDNA for diagnostic digestion (see panel C) are shown. **(B)** Timeline showing transfection of RHΔku80Δhxgprt parasites with the Cas9 GFP guide vector and ds repair template on day 0, sorting of eGFP+ parasites on day 2, and diagnostic digest of gDNA from individual eGFP+ clones on day 9. **(C)** Left schematic shows predicted fragment sizes after BglII digestion of the PCR fragment generated by P5/P11, for parental (WT) and T130A parasites. Right panel shows the BglII restriction digest of a T130A positive clone (leftmost lane) and WT clone (middle lane); the 201 and 88 bp fragments are not resolved on this gel. Right lane = ladder, selected band sizes indicated in kb. Clones that showed the expected digestion pattern for the T130A substitution were verified by sequencing using primers P5-P13 (S1 Table). The original uncropped DNA gel from which panel C is derived is shown in S1 Raw Images. (TIF)

**S11 Fig. The TgMyoA T130A mutation has little to no impact on parasite growth in vitro or virulence in vivo. (A)** Representative images of a plaque assay using wild-type (top image; WT) and T130A mutant (bottom image) parasites, 7 days after inoculating confluent HFF monolayers with 100 parasites/well. The entire well (22.6 mm diameter) from the 12-well plate is shown for each parasite line; scale bars = 4 mm. **(B)** Total number of plaques per well and average plaque area in mm$^2$ of HFF monolayers inoculated with 10, 100, 200, and 500 parasites. The data represent the mean ± SEM from 4 independent biological replicates. There was no significant difference in plaque number or size between WT and T130A parasites at any of the inoculum sizes assayed. **(C)** Nine mice were infected on day 0 with 50 (top panel) or 1,000 (bottom panel) wild-type RH (black circles) or T130A mutant parasites (orange squares). There was no significant difference in survival over the next 11 days between mice infected with the same number of wild-type or T130A parasites (log-rank Mantel–Cox test). The measurements underlying the data plotted in panels B and C can be found in S16 and S17 Data, respectively. (TIF)

**S12 Fig. Parasite displacement in the 3D motility assay is less impacted by KNX-002 treatment in parasites expressing the TgMyoA T130A mutation than in wild-type parasites.** Displacement of wild-type (WT) and T130A mutant parasites during 80 s of motility in Matrigel, in the presence of 0 (DMSO vehicle only), 10, or 20 μM KNX-002. The data were derived from the same set of experiments shown in Fig 7B–7D. Each data point represents a single biological replicate composed of 3 technical replicates; 3 of the biological replicates were collected on the same 3 days (circles), and 3 of the biological replicates were collected on a different 3 days (squares). Bars show the mean of the biological replicates ± SEM. Sets of biological replicates using the same parasite line and collected on the same days were compared by Student's one-tailed paired *t* tests (significance indicated above the graphs). Sets of biological replicates comparing different parasite lines were analyzed by Student's two-tailed unpaired *t* tests (significance indicated below the graphs). ns = not significant. The measurements underlying the

data plotted in this figure can be found in S18 and S19 Data.
(TIF)

**S13 Fig. KNX-002 lowers the duty ratio of the TgMyoA motor.** Actin filament velocity (V/Vmax) was plotted as a function of the number of myosin heads capable of interacting with the actin filament. The duty ratios calculated from these data (see Methods) were significantly different for TgMyoA treated with DMSO (gray datapoints) vs. 10 μM KNX-002 (orange; Kolmogorov–Smirnov test, $p < 0.0001$). The measurements underlying the calculated duty ratios shown in this figure can be found in S22 Data.
(TIF)

**S14 Fig. The T130A mutation neither up-regulates TgMyoA nor changes the expression of other known motility-related genes. (A)** The expression level of TgMyoA in wild-type (WT) and T130A mutant parasites was compared via quantitative western blotting. The blot was also probed for TgACT1 as a loading control; when normalized to the corresponding actin signals, the TgMyoA band intensities were within 7% of each other. **(B)** Left panel: The most differentially expressed genes between WT (RHΔku80) and T130A parasites (20 up-regulated and 10 down-regulated), after filtering for padj < 0.05 and FPKM > 0.1 and eliminating any genes that did not show the same pattern in all 3 replicates. Right panel: Analysis of differential expression of selected genes with the potential to compensate for loss of TgMyoA function, including other myosin motors, known MyoA-interacting proteins, glideosome components, actin and actin regulatory factors. The original uncropped western blot from which panel A is derived is shown in S1 Raw Images and the original expression data from which the highlighted results in panel B are derived are found in S23 Data.
(TIF)

**S1 Table. Primers and oligonucleotides used in this study.**
(TIF)

**S1 Raw Images. Original uncropped, unprocessed DNA gel from which S10C Fig was derived and original uncropped, unprocessed western blot from which S14A Fig was derived.**
(PDF)

**S1 Data. Actin-activated ATPase activity of TgMyoA and vertebrate myosins in the presence of different amounts of KNX-002, relative to the activity in the presence of an equivalent amount of DMSO.** Each number corresponds to the mean activity of a single biological replicate comprised of 2 technical replicates run at the same time on 2 different plates.
(XLSX)

**S2 Data. Effects of KNX-002 on *T. gondii* growth in culture and host cell viability/toxicity.** Tab A shows raw growth assay data, expressed in relative fluorescence units [RFU], of parasites grown in the presence of differing concentrations of KNX-002. Tab B shows the day 5 growth assay data expressed relative to average growth in an equivalent amount of DMSO; these numbers were used to calculate the compound's $IC_{50}$. The data in Tabs C and D show the results from CellTiter-Glo (panel C) and CellTox green (panel D) assays on HepG2 and HFF cells in the presence of 0 (DMSO vehicle only), 40, or 80 μM KNX-002.
(XLSX)

**S3 Data. Comparison of the effects of DMSO vs. 80 μM KNX-002 on the replication of *T. gondii* in culture, measured by the number of parasite per vacuole at 12, 24, and 36 h post-infection.** Each number represents the mean from a single biological replicate consisting of 2

technical replicates.
(XLSX)

**S4 Data. Effects of 20 μM KNX-002 and its analogs on TgMyoA ATPAase activity, expressed in relative luminescence units (RLU).**
(XLSX)

**S5 Data. Effects of different concentrations (0–80 μM) of KNX-002 and its analogs on parasite growth, expressed in relative fluorescence units (RFU).**
(XLSX)

**S6 Data. Growth of parasites in different concentrations (0–80 μM) of KNX-002 and its analogs, relative to average growth in the presence of DMSO, calculated from the individual technical replicate measurements in S5 Data.** These relative growth numbers were used to calculate the $IC_{50}$ for each compound on the indicated day postinfection. For the compounds that did not show sufficient inhibition from 0–80 μM to determine an $IC_{50}$, the relative growth numbers from both day 5 and day 6 are shown.
(XLSX)

**S7 Data. Effects of KNX-002 and its analogs (at 40 μM or 80 μM) vs. DMSO on the viability of (CellTiter-Glo) and toxicity towards (CellTox green) HepG2 and HFF cells.** Each number represents the mean of a single biological replicate consisting of 2–4 technical replicates. Where indicated, 0.1% w/v sodium azide was used as a positive control for toxicity.
(XLSX)

**S8 Data Tab A. Viability of parasites incubated for the indicated time in DMSO, 40 μM or 80 μM VEST5, as measured using the CellTiter-Glo assay.** Numbers for **VEST5** indicate % luminescence compared to the average of the DMSO controls. Each number represents the mean measurement from a single biological replicate consisting of 3 technical replicates. Tab B. Comparison of the effects of DMSO vs. 80 μM **VEST5** on the replication of *T. gondii* in culture, measured by quantifying the number of parasites per vacuole at 12, 24, and 36 h postinfection. Each number represents the mean percentage measured in a single biological replicate consisting of 2 technical replicates (≥50 vacuoles each).
(XLSX)

**S9 Data. Raw *xyz* positional information over time for all wild-type (WT) and T130A mutant parasites showing displacements >2 μm in the presence of either DMSO (vehicle) or the indicated concentrations of KNX-002.**
(XLSX)

**S10 Data. Motility parameters calculated from the positional information in S9 Data (see Methods for details).**
(XLSX)

**S11 Data. Effects of different concentrations of KNX-002 on the growth of wild-type parasites and the 5 clonal mutants showing reduced sensitivity (>2.5-fold) to the compound, expressed in relative fluorescence units (RFU).**
(XLSX)

**S12 Data. Relative growth of wild-type parasites and 5 clones showing reduced sensitivity to KNX-002, in the presence of different concentrations of the compound.** Growth relative to average DMSO was calculated on day 5 postinfection from the measurements in S11 Data; these growth numbers were used to calculate the $IC_{50}$ for KNX-002 in each of the parasite

lines.
(XLSX)

**S13 Data. Speed of individual actin filaments (μm/sec) in the in vitro motility assay, using either wild-type (WT) or T130A TgMyoA and in the presence of different concentrations of KNX-002.**
(XLSX)

**S14 Data. Absorbance measurements (A280; mAU) of wild-type and T130A TgMyoA (residues 1–778) eluting from a Superdex SX-75 size exclusion column.**
(XLSX)

**S15 Data. Differential scanning fluorimetry results showing the fluorescence recorded at different temperatures for residues 1–778 of wild-type (wells A04-A06) and T130A (wells B04-06) TgMyoA.** The derivative fluorescence calculated from these data is also shown, beginning at row 2491.
(CSV)

**S16 Data. Number and area of plaques formed 7 days after infection of an HFF monolayer with different numbers of wild-type (WT) or T130A mutant parasites.**
(XLSX)

**S17 Data. Percentage of mice surviving over time after infection with either 50 or 1,000 wild-type (RH) or T130A mutant parasites; 9 mice per group.**
(XLSX)

**S18 Data. Raw *xyz* positional information over time for all wild-type (WT) and T130A mutant parasites showing displacements >2 μm in the presence of DMSO (vehicle), 10 μM KNX-002, or 20 μM KNX-002.**
(XLSX)

**S19 Data. Motility parameters calculated from the positional information in S18 Data (see Methods for details).**
(XLSX)

**S20 Data. Number and area of plaques formed 7 days after infection of an HFF monolayer with different numbers of wild-type (WT) or T130A mutant parasites, in the absence (DMSO vehicle only; Tab A) or presence of varying amounts of KNX-002 (Tabs B–D).**
(XLSX)

**S21 Data. Percentage of mice surviving over time after infection with 500 wild-type (RH) or T130A mutant parasites and treatment with 20 mg/kg KNX-002 or an equivalent volume of DMSO (vehicle); 10 mice per group.**
(XLSX)

**S22 Data. Individual actin filament lengths and speeds in in vitro motility assays carried out in the absence (DMSO vehicle only) or presence of 10 μM KNX-002 and using different dilutions of TgMyoA to coat the coverslip, as indicated (undiluted TgMyoA = 0.02 mg/ml).**
(XLSX)

**S23 Data. RNAseq data comparing differential gene expression in WT (RHΔku80) and T130A parasites; the data from 3 biological replicates are shown.**
(XLSM)

**S1 Text. Compound synthesis and analysis.**
(PDF)

## Acknowledgments

We thank Cytokinetics, for their willingness to provide staff time and reagents to this project, Samantha Previs for technical support, Dr. Shane Nelson for help with the duty ratio analysis, and Dr. Roxanna del Rio-Guerra in the UVM Larner College of Medicine Harry Hood Basset Flow Cytometry and Cell Sorting Facility (RRID: SSR_022147) for advice and assistance with cell sorting. Genomic-scale data sets and ancillary information were obtained from the Toxoplasma Genome Database (ToxoDB.org). ToxoDB is a component of the Eukaryotic Pathogen, Vector and Host Informatics Resource [79], supported by the National Institutes of Allergy and Infectious Diseases and the Wellcome Trust; we gratefully acknowledge the staff responsible for developing and maintaining this resource.

## Author Contributions

**Conceptualization:** Anne Kelsen, Robyn S. Kent, Anne K. Snyder, Eddie Wehri, Rachel V. Stadler, Bruno Martorelli di Genova, Martin J. Boulanger, David M. Warshaw, Nicholas J. Westwood, Julia Schaletzky, Gary E. Ward.

**Data curation:** Anne Kelsen, Robyn S. Kent, Anne K. Snyder, Eddie Wehri, Stephen J. Bishop, Rachel V. Stadler, Cameron Powell, Martin J. Boulanger, Nicholas J. Westwood, Julia Schaletzky, Gary E. Ward.

**Formal analysis:** Anne Kelsen, Robyn S. Kent, Anne K. Snyder, Eddie Wehri, Stephen J. Bishop, Rachel V. Stadler, Cameron Powell, Pramod K. Rompikuntal, Martin J. Boulanger, David M. Warshaw, Nicholas J. Westwood, Julia Schaletzky, Gary E. Ward.

**Funding acquisition:** Martin J. Boulanger, David M. Warshaw, Nicholas J. Westwood, Julia Schaletzky, Gary E. Ward.

**Investigation:** Anne Kelsen, Robyn S. Kent, Anne K. Snyder, Eddie Wehri, Stephen J. Bishop, Rachel V. Stadler, Cameron Powell, Bruno Martorelli di Genova, Pramod K. Rompikuntal, Martin J. Boulanger, David M. Warshaw, Nicholas J. Westwood, Julia Schaletzky, Gary E. Ward.

**Methodology:** Anne Kelsen, Robyn S. Kent, Anne K. Snyder, Eddie Wehri, Stephen J. Bishop, Rachel V. Stadler, Cameron Powell, Bruno Martorelli di Genova, Pramod K. Rompikuntal, Martin J. Boulanger, David M. Warshaw, Nicholas J. Westwood, Julia Schaletzky, Gary E. Ward.

**Project administration:** Martin J. Boulanger, David M. Warshaw, Nicholas J. Westwood, Julia Schaletzky, Gary E. Ward.

**Resources:** Martin J. Boulanger, David M. Warshaw, Nicholas J. Westwood, Julia Schaletzky, Gary E. Ward.

**Supervision:** Robyn S. Kent, Bruno Martorelli di Genova, Martin J. Boulanger, David M. Warshaw, Nicholas J. Westwood, Julia Schaletzky, Gary E. Ward.

**Validation:** Anne Kelsen, Robyn S. Kent, Anne K. Snyder, Eddie Wehri, Stephen J. Bishop, Rachel V. Stadler, Cameron Powell, Bruno Martorelli di Genova, Pramod K. Rompikuntal, Martin J. Boulanger, David M. Warshaw, Nicholas J. Westwood, Julia Schaletzky, Gary E. Ward.

**Visualization:** Anne Kelsen, Robyn S. Kent, Anne K. Snyder, Eddie Wehri, Stephen J. Bishop, Rachel V. Stadler, Cameron Powell, Pramod K. Rompikuntal, Martin J. Boulanger, David M. Warshaw, Nicholas J. Westwood, Julia Schaletzky, Gary E. Ward.

**Writing – original draft:** Anne Kelsen, Robyn S. Kent, Anne K. Snyder, Eddie Wehri, Stephen J. Bishop, Martin J. Boulanger, David M. Warshaw, Nicholas J. Westwood, Julia Schaletzky, Gary E. Ward.

**Writing – review & editing:** Anne Kelsen, Robyn S. Kent, Anne K. Snyder, Eddie Wehri, Stephen J. Bishop, Rachel V. Stadler, Cameron Powell, Bruno Martorelli di Genova, Pramod K. Rompikuntal, Martin J. Boulanger, David M. Warshaw, Nicholas J. Westwood, Julia Schaletzky, Gary E. Ward.

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
