## [Editor Report · Decision Letter 0]

3 Mar 2023

Dear Dr. Ward, 

Thank you for submitting your manuscript entitled "The class XIV myosin of Toxoplasma gondii, TgMyoA, is druggable in an animal model of infection" for consideration as a Research Article by PLOS Biology.

Your manuscript, the reviewers reports from Review Commons, and your revision has now been evaluated by the PLOS Biology editorial staff, as well as by an academic editor with relevant expertise, and I am writing to let you know that we would like to consider further your manuscript for publication. We have decided that further peer review is not necessary, nor further experimental revision. However, we consider that a minor revision is necessary and some policy and editorial issues that need to be addressed. 

However, before we can send you a detailed decision letter, we need you to complete your submission by providing the metadata. To this end, please login to Editorial Manager where you will find the paper in the 'Submissions Needing Revisions' folder on your homepage. Please click 'Revise Submission' from the Action Links and complete all additional questions in the submission questionnaire.

Once your full submission is complete, your paper will undergo a series of checks. After your manuscript has passed the checks, we will send you the detailed decision letter. To provide the metadata for your submission, please Login to Editorial Manager (https://www.editorialmanager.com/pbiology) within two working days, i.e. by Mar 05 2023 11:59PM.

Kind regards,

Paula

---

Senior Editor

PLOS Biology

---

## [Editor Report · Decision Letter 1]

7 Mar 2023

Dear Dr. Ward,

Thank you for your patience while we considered your manuscript "The class XIV myosin of Toxoplasma gondii, TgMyoA, is druggable in an animal model of infection" for publication as a Research Article at PLOS Biology. Your manuscript, the comments from the Review Commons' reviewers, and your revision have been evaluated by the PLOS Biology editors and an Academic Editor with relevant expertise.

Based on our Academic Editor's assessment of your revision, we are likely to accept this manuscript for publication, provided you satisfactorily address the comments from the Academic Editor that you can find at the end of this letter.

In particular, it is important that you tone down the interpretation of your results, including the use of enhancers, and that you modify the title and abstract.

Please also make sure to address the following data and other policy-related requests.

1. DATA POLICY:

A) Supplementary files (e.g., excel). Please ensure that all data files are uploaded as 'Supporting Information' and are invariably referred to (in the manuscript, figure legends, and the Description field when uploading your files) using the following format verbatim: S1 Data, S2 Data, etc. Multiple panels of a single or even several figures can be included as multiple sheets in one excel file that is saved using exactly the following convention: S1_Data.xlsx (using an underscore).

B) Deposition in a publicly available repository. Please also provide the accession code or a reviewer link so that we may view your data before publication.

Regardless of the method selected, please ensure that you provide the individual numerical values that underlie the summary data displayed in the following figure panels as they are essential for readers to assess your analysis and to reproduce it: Figures 1A, 2ABCD, 4C, 5AB, 6AB, 7BCD, 8B, 9, and Supplementary Figures SF1, SF2, SF3, SF4AB, SF5, SF6AB, SF7AB, SF8AB, SF9AB, SF11BC, SF12, SF13, SF14B.

**Please also ensure that figure legends in your manuscript include information on where the underlying data can be found, and ensure your supplemental data file/s has a legend.**

We require the original, uncropped and minimally adjusted images supporting all blot and gel results reported in an article's figures or Supporting Information files. We will require these files before a manuscript can be accepted so please prepare and upload them now. We will need this for Supplementary Figures SF10C, SF14A.

3. Please add size bars to the microscopy pictures in Figures 8A and SF11A.

4. Please simplify the title and eliminate the reference to the animals, as suggested by the Academic Editor. We suggest: "Inhibition of Toxoplasma gondii myosin A as an effective strategy for toxoplasmosis".

Please carefully read our guidelines for how to prepare and upload this data: https://journals.plos.org/plosbiology/s/figures#loc-blot-and-gel-reporting-requirements

We expect to receive your revised manuscript within two weeks.

*Published Peer Review History*

*Press*

Sincerely,

Paula

---

Senior Editor,

pjaureguionieva@plos.org,

PLOS Biology

EDITED COMMENTS FROM THE ACADEMIC EDITOR:

I suggest the authors to make less use of emotional enhancers like dramatically, marked, clearly … just stick to describing what was found and let the reader weigh the evidence herself.

Second is the title. This is something the reviewers brought up. I feel the word druggable is fine - that is indeed what the bulk of the paper shows - however I would remove the mention of animals in the title. This is a great paper showing that MyoA can be inhibited and it provides new chemical matter to do so, resistance mutants, a bunch of biochemical and cell biological assays - all nicely done and rigorous. The mouse data is a nice add on at the end, but also the weakest part of the study. It’s a single experiment that used infrequent dosing due to toxicity and the in vivo potency is not great … most of the mice still die. On its own this experiment does not say so much about the prospect of MyoA as a target. The reviewers offered two avenues: provide more mouse data and show that this is a good drug in vivo (which is difficult and not a bar that the authors should be held to for acceptance here) or t take it out of the title and tone down the abstract a bit (the discussion actually does an entirely fair job at discussing in vivo caveats and the need for further SAR and compound optimization).

276 results in a markedly different motility pattern, consisting almost entirely of small diameter circles, and 25 μM KNX-002 dramatically inhibits the number of parasites undergoing any movement at all ( … at 25 uM most parasites do not move at all ….)

The authors use the phrase ‘marked’ contrast frequently, at times when the differences are more moderate, e.g. in the mouse experiments where its’s used twice. I suggest to go through the manuscript and to remove some of these enhancers and simply state what that there is difference and quantify that difference.

536 cut the in front of cilia

677 cut ‘a’ in front of few

---

## [Editor Report · Decision Letter 2]

5 Apr 2023

Dear Dr. Ward,

Thank you for the submission of your revised Research Article "MyosinA is a druggable target in the widespread protozoan parasite Toxoplasma gondii" for publication in PLOS Biology. On behalf of my colleagues and the Academic Editor, Boris Striepen, I am pleased to say that we can in principle accept your manuscript for publication, provided you address any remaining formatting and reporting issues. These will be detailed in an email you should receive within 2-3 business days from our colleagues in the journal operations team; no action is required from you until then. Please note that we will not be able to formally accept your manuscript and schedule it for publication until you have completed any requested changes.

PRESS

Sincerely, 

Paula 

---

Senior Editor

PLOS Biology
